# Towards A Unified Min-Max Framework for Adversarial Exploration and Robustness

## Abstract

The worst-case training principle that minimizes the maximal adversarial loss, also known as adversarial training (AT), has shown to be a state-of-the-art approach for enhancing adversarial robustness against norm-ball bounded input perturbations. Nonetheless, min-max optimization beyond the purpose of AT has not been rigorously explored in the research of adversarial attack and defense. In particular, given a set of risk sources (domains), minimizing the maximal loss induced from the domain set can be reformulated as a general min-max problem that is fundamentally different from AT since the maximization is taken over the probability simplex of the domain set. Examples of this general formulation include attacking model ensembles, devising universal perturbation under multiple inputs or data transformations, and generalized AT over different types of attack models. We show that these problems can be solved under a unified and theoretically principled min-max optimization framework. We also show that the self-adjusted domain weights learnt from our method provide a holistic tool to explain the difficulty level of attack and defense over multiple domains. Extensive experiments show that our approach leads to substantial performance improvement over the conventional heuristic strategies[1].

## 1 Introduction

Training a machine learning model that is capable of assuring its worst-case performance against all possible adversaries given a specified threat model is a fundamental yet challenging problem, especially for deep neural networks (DNNs) (Szegedy et al., 2013; Goodfellow et al., 2015; Carlini & Wagner, 2017). A common practice to train an adversarially robust model is based on a specific form of min-max training, known as *adversarial training* (AT) (Goodfellow et al., 2015; Madry et al., 2017), where the minimization step learns model weights under the adversarial loss constructed at the maximization step in an alternative training fashion. On datasets such as MNIST and CIFAR-10, AT has achieved the state-of-the-art defense performance against $\ell_p$-norm-ball input perturbations (Athalye et al., 2018b).

Motivated by the success of AT, one follow-up question that naturally arises is: *Beyond AT, can other types of min-max formulation and optimization techniques advance the research in adversarial robustness?* In this paper, we give an affirmative answer corroborated by the substantial performance gain and the ability of self-learned risk interpretation using our proposed min-max framework on several tasks for adversarial attack and defense.

We demonstrate the utility of a general formulation for min-max optimization minimizing the maximal loss induced from a set of risk sources (domains). Our considered min-max formulation is fundamentally different from AT, as our maximization step is taken over the probability simplex of the set of domains. Moreover, we show that many problem setups in adversarial attacks and defenses can in fact be reformulated under this general min-max framework, including attacking model ensembles (Tramèr et al., 2018; Liu et al., 2018), devising universal perturbation to input samples (Moosavi-Dezfooli et al., 2017) or data transformations (Athalye & Sutskever, 2018; Brown et al., 2017), and generalized AT over multiple types of threat models (Tramèr & Boneh, 2019; Araujo et al., 2019). However, current methods for solving these tasks often rely on simple heuristics (e.g.,

---

[1]For reproducibility, the code and trained models will be released accompanying this paper.

uniform averaging), resulting in significant performance drops when comparing to our proposed min-max optimization framework.

Specifically, based on the general min-max framework, we show that these problems can be solved under the same optimization procedure and prove the rate of its algorithmic convergence. As a byproduct and an exclusive feature, by tracking the weighting factor associated with the probability simplex during training, our method can provide tools for self-adjusted risk assessment and obtain novel insights on the set of domains for the associated tasks.

**Contributions**  (1) We indicate the utility of min-max optimization beyond AT by proposing a general and theoretically grounded framework on adversarial attack and defense. (2) We demonstrate the effectiveness of our min-max framework by evaluating the proposed APGD attack on MNIST and CIFAR-10. In theory, we show that APGD has an $O(1/T)$ convergence rate, where $T$ is the number of iterations. In practice, we show that APGD obtains 17.48%, 35.21% and 9.39% improvement on average compared with PGD attack on CIFAR-10. (3) We propose a generalized AT scheme under mixed types of adversarial attacks and demonstrate that the diversified attack ensemble helps adversarial robustness. Compared with vanilla AT, our new training scheme leads to better worst-case robustness even if the defender lacks prior knowledge of the strengths of attacks. (4) We show how the weighting factors of the probability simplex help to obtain novel insights for associated tasks and interpreting the importance of candidates in domains.

**Related Work**  Recent studies have identified that DNNs are highly vulnerable to adversarial manipulations in various applications (Szegedy et al., 2013; Carlini et al., 2016; Jia & Liang, 2017; Lin et al., 2017; Huang et al., 2017; Carlini & Wagner, 2018; Zhao et al., 2018; Eykholt et al., 2018; Chen et al., 2018a; Lei et al., 2019), thus leading to an arms race between adversarial attacks (Carlini & Wagner, 2017; Athalye et al., 2018b; Goodfellow et al., 2014; Papernot et al., 2016a; Moosavi Dezfooli et al., 2016; Chen et al., 2018b; Xu et al., 2019) and defenses (Madry et al., 2017; Papernot et al., 2016b; Meng & Chen, 2017; Xie et al., 2017; Xu et al., 2018). One intriguing property of adversarial examples is the transferability across multiple domains (Liu et al., 2017; Tramèr et al., 2017; Papernot et al., 2017; Su et al., 2018), which indicates a more challenging yet promising research direction – devising universal adversarial perturbations over model ensembles (Tramèr et al., 2018; Liu et al., 2018), input samples (Moosavi-Dezfooli et al., 2017; Metzen et al., 2017; Shafahi et al., 2018) and data transformations (Athalye et al., 2018b; Athalye & Sutskever, 2018; Brown et al., 2017). However, current approaches suffer from a significant performance loss for resting on the uniform averaging strategy. We will compare these works with our min-max method in Sec. 4. As a natural extension following min-max attack, we study the generalized AT under multiple perturbations (Tramèr & Boneh, 2019; Araujo et al., 2019; Kang et al., 2019; Croce & Hein, 2019). Finally, our min-max framework is adapted and inspired by previous literature on robust learning over multiple domains (Qian et al., 2018; Rafique et al., 2018; Lu et al., 2018; 2019a).

## 2 MIN-MAX POWER IN ADVERSARIAL EXPLORATION AND ROBUSTNESS

We begin by introducing the principle of robust learning over multiple domains and its connection to a specialized form of min-max optimization. We then show that the resulting min-max formulation fits into various attack settings for *adversarial exploration*: a) ensemble adversarial attack, b) universal adversarial perturbation and c) robust perturbation over data transformations. Finally, we propose a generalized adversarial training (AT) framework under mixed types of adversarial attacks to improve *model robustness*.

### 2.1 GENERAL IDEA: ROBUST LEARNING OVER MULTIPLE DOMAINS

Consider $K$ loss functions $\{F_i(\mathbf{v})\}$ (each of which is defined on a learning domain), the problem of robust learning over $K$ domains can be formulated as (Qian et al., 2018; Rafique et al., 2018; Lu et al., 2018)

$$\underset{\mathbf{v} \in \mathcal{V}}{\text{minimize}} \; \underset{\mathbf{w} \in \mathcal{P}}{\text{maximize}} \quad \sum_{i=1}^{K} w_i F_i(\mathbf{v}), \tag{1}$$

where $\mathbf{v}$ and $\mathbf{w}$ are optimization variables, $\mathcal{V}$ is a constraint set, and $\mathcal{P}$ denotes the probability simplex $\mathcal{P} = \{\mathbf{w} \,|\, \mathbf{1}^T \mathbf{w} = 1, w_i \in [0, 1], \forall i\}$. Since the inner maximization problem in (1) is a linear function

of $\mathbf{w}$ over the probabilistic simplex, problem (1) is thus equivalent to

$$\underset{\mathbf{v}\in\mathcal{V}}{\text{minimize}}\ \underset{i\in[K]}{\text{maximize}}\quad F_i(\mathbf{v}), \tag{2}$$

where $[K]$ denotes the integer set $\{1, 2, \ldots, K\}$.

**Benefit and computation challenge of** (1)   Compared to multi-task learning in a finite-sum formulation which minimizes $K$ losses on *average*, problem (1) provides consistently robust *worst-case* performance across all domains. This can be explained from the epigraph form of (2),

$$\underset{\mathbf{v}\in\mathcal{V},t}{\text{minimize}}\ t,\quad \text{subject to } F_i(\mathbf{v}) \le t, i \in [K], \tag{3}$$

where $t$ is an epigraph variable (Boyd & Vandenberghe, 2004) that provides the $t$-level robustness at each domain.

Although the min-max problem (1) offers a great robustness interpretation as in (3), solving it becomes more challenging than solving the finite-sum problem. It is clear from (2) that the inner maximization problem of (1) always returns the one-hot value of $\mathbf{w}$, namely, $\mathbf{w} = \mathbf{e}_i$, where $\mathbf{e}_i$ is the $i$th standard basis vector, and $i = \arg\max_i\{F_i(\mathbf{v})\}$. The one-hot coding reduces the generalizability to other domains and induces instability of the learning procedure in practice. Such an issue is often mitigated by introducing a strongly concave regularizer in the inner maximization step (Lu et al., 2018; Qian et al., 2018).

**Regularized problem formulation**   Spurred by (Qian et al., 2018), we penalize the distance between the *worst-case* loss and the *average* loss over $K$ domains. This yields

$$\underset{\mathbf{v}\in\mathcal{V}}{\text{minimize}}\ \underset{\mathbf{w}\in\mathcal{P}}{\text{maximize}}\quad \sum_{i=1}^{K} w_i F_i(\mathbf{v}) - \frac{\gamma}{2}\|\mathbf{w} - \mathbf{1}/K\|_2^2, \tag{4}$$

where $\gamma > 0$ is a regularization parameter. As $\gamma \to 0$, problem (4) is equivalent to (1). By contrast, it becomes the finite-sum problem when $\gamma \to \infty$ since $\mathbf{w} \to \mathbf{1}/K$. *In this sense, the trainable $\mathbf{w}$ provides an essential indicator on the importance level of each domain.* The larger the weight is, the more important the domain is. We call $\mathbf{w}$ *domain weights* in this paper. We next show how the principle of robust learning over multiple domains can fit into various settings of adversarial attack and defense problems.

## 2.2   Robust adversarial attacks

The general goal of adversarial attack is to craft an adversarial example $\mathbf{x}' = \mathbf{x}_0 + \boldsymbol{\delta} \in \mathbb{R}^d$ to mislead the prediction of machine learning (ML) or deep learning (DL) systems, where $\mathbf{x}_0$ denotes the natural example with the true label $t_0$, and $\boldsymbol{\delta}$ is known as *adversarial perturbation*, commonly subject to $\ell_p$-norm ($p \in \{0, 1, 2, \infty\}$) constraint $\mathcal{X} := \{\boldsymbol{\delta} \,|\, \|\boldsymbol{\delta}\|_p \le \epsilon,\ \mathbf{x}_0 + \boldsymbol{\delta} \in [0,1]^d\}$ for a given small number $\epsilon$. Here the $\ell_p$ norm enforces the similarity between $\mathbf{x}'$ and $\mathbf{x}_0$, and the input space of ML/DL systems is normalized to $[0, 1]^d$.

**Ensemble attack over multiple models**   Consider $K$ ML/DL models $\{\mathcal{M}_i\}_{i=1}^K$, the goal is to find robust adversarial examples that can fool all $K$ models simultaneously. In this case, the notion of 'domain' in (4) is specified as 'model', and the objective function $F_i$ in (4) signifies the attack loss $f(\boldsymbol{\delta}; \mathbf{x}_0, y_0, \mathcal{M}_i)$ given the natural input $(\mathbf{x}_0, y_0)$ and the model $\mathcal{M}_i$. Thus, problem (4) becomes

$$\underset{\boldsymbol{\delta}\in\mathcal{X}}{\text{minimize}}\ \underset{\mathbf{w}\in\mathcal{P}}{\text{maximize}}\quad \sum_{i=1}^{K} w_i f(\boldsymbol{\delta}; \mathbf{x}_0, y_0, \mathcal{M}_i) - \frac{\gamma}{2}\|\mathbf{w} - \mathbf{1}/K\|_2^2, \tag{5}$$

where $\mathbf{w}$ encodes the difficulty level of attacking each model.

**Universal perturbation over multiple examples**   Consider $K$ natural examples $\{(\mathbf{x}_i, y_i)\}_{i=1}^K$ and a single model $\mathcal{M}$, our goal is to find the universal perturbation $\boldsymbol{\delta}$ so that all the corrupted $K$ examples can fool $\mathcal{M}$. In this case, the notion of 'domain' in (4) is specified as 'example', and problem (4) becomes

$$\underset{\boldsymbol{\delta}\in\mathcal{X}}{\text{minimize}}\ \underset{\mathbf{w}\in\mathcal{P}}{\text{maximize}}\quad \sum_{i=1}^{K} w_i f(\boldsymbol{\delta}; \mathbf{x}_i, y_i, \mathcal{M}) - \frac{\gamma}{2}\|\mathbf{w} - \mathbf{1}/K\|_2^2, \tag{6}$$

where different from (5), $\mathbf{w}$ encodes the difficulty level of attacking each example.

**Adversarial attack over data transformations** Consider $K$ categories of data transformation $\{p_i\}$, e.g., rotation, lightening, and translation (Athalye et al., 2018a), our goal is to find the adversarial attack that is robust to data transformations. In this case, the notion of 'domain' in (4) is specified as 'data transformer', and problem (4) becomes

$$\underset{\boldsymbol{\delta} \in \mathcal{X}}{\text{minimize}} \ \underset{\mathbf{w} \in \mathcal{P}}{\text{maximize}} \quad \sum_{i=1}^{K} w_i \mathbb{E}_{t \sim p_i}[f(t(\mathbf{x}_0 + \boldsymbol{\delta}); y_0, \mathcal{M})] - \frac{\gamma}{2}\|\mathbf{w} - \mathbf{1}/K\|_2^2, \tag{7}$$

where $\mathbb{E}_{t \sim p_i}[f(t(\mathbf{x}_0 + \boldsymbol{\delta}); y_0, \mathcal{M})]$ denotes the attack loss under the distribution of data transformation $p_i$, and $\mathbf{w}$ encodes the difficulty level of attacking each type of transformed example $\mathbf{x}_0$.

## 2.3 Adversarial training (AT) under mixed types of adversarial attacks

Conventional AT is restricted to a single type of norm-ball constrained adversarial attack (Madry et al., 2017). For example, AT under $\ell_\infty$ attack yields

$$\underset{\boldsymbol{\theta}}{\text{minimize}} \ \mathbb{E}_{(\mathbf{x}, \mathbf{y}) \in \mathcal{D}} \ \underset{\|\boldsymbol{\delta}\|_\infty \leq \epsilon}{\text{maximize}} \ f_{\text{tr}}(\boldsymbol{\theta}, \boldsymbol{\delta}; \mathbf{x}, y), \tag{8}$$

where $\boldsymbol{\theta} \in \mathbb{R}^n$ denotes model parameters, $\boldsymbol{\delta}$ denotes $\epsilon$-tolerant $\ell_\infty$ attack, and $f_{\text{tr}}(\boldsymbol{\theta}, \boldsymbol{\delta}; \mathbf{x}, y)$ is the training loss under perturbed examples $\{(\mathbf{x} + \boldsymbol{\delta}, y)\}$. However, there possibly exist blind attacking spots across multiple types of adversarial attacks so that AT under one attack would not be strong enough against another attack (Araujo et al., 2019). Thus, an interesting question is how to generalize AT under multiple types of adversarial attacks. One possible way is to use the finite-sum formulation

$$\underset{\boldsymbol{\theta}}{\text{minimize}} \ \mathbb{E}_{(\mathbf{x}, \mathbf{y}) \in \mathcal{D}} \ \underset{\{\boldsymbol{\delta}_i \in \mathcal{X}_i\}}{\text{maximize}} \ \frac{1}{K}\sum_{i=1}^{K} f_{\text{tr}}(\boldsymbol{\theta}, \boldsymbol{\delta}_i; \mathbf{x}, y), \tag{9}$$

where $\boldsymbol{\delta}_i \in \mathcal{X}_i$ is the $i$th type of adversarial perturbation defined on $\mathcal{X}_i$, e.g., different $\ell_p$ attacks.

Moreover, one can map 'attack type' to 'domain' considered in (1). We then perform AT against the *strongest* adversarial attack across $K$ attack types in order to avoid blind attacking spots. That is, upon defining $F_i(\boldsymbol{\theta}) := \text{maximize}_{\boldsymbol{\delta}_i \in \mathcal{X}_i} \ f_{\text{tr}}(\boldsymbol{\theta}, \boldsymbol{\delta}_i; \mathbf{x}, y)$, we solve the problem of the form (2),

$$\underset{\boldsymbol{\theta}}{\text{minimize}} \ \mathbb{E}_{(\mathbf{x}, \mathbf{y}) \in \mathcal{D}} \ \underset{i \in [K]}{\text{maximize}} \ F_i(\boldsymbol{\theta}). \tag{10}$$

In fact, problem (10) is in the min-max-max form, however, Lemma 1 shows that problem (10) can be further simplified to the min-max form.

**Lemma 1.** *Problem* (10) *is equivalent to*

$$\underset{\boldsymbol{\theta}}{\text{minimize}} \ \mathbb{E}_{(\mathbf{x}, \mathbf{y}) \in \mathcal{D}} \ \underset{\mathbf{w} \in \mathcal{P}, \{\boldsymbol{\delta}_i \in \mathcal{X}_i\}}{\text{maximize}} \ \sum_{i=1}^{K} w_i f_{\text{tr}}(\boldsymbol{\theta}, \boldsymbol{\delta}_i; \mathbf{x}, y), \tag{11}$$

*where* $\mathbf{w} \in \mathbb{R}^K$ *represent domain weights, and* $\mathcal{P}$ *has been defined in* (1).

**Proof**: see Appendix A.

Similar to (4), a strongly concave regularizer $-\gamma/2\|\mathbf{w} - \mathbf{1}/K\|_2^2$ can be added into the inner maximization problem of (11), which can boost the stability of the learning procedure and strike a balance between the max and the average attack performance. However, solving problem (11) and its regularized version is more complicated than (8) since the inner maximization involves both domain weights $\mathbf{w}$ and adversarial perturbations $\{\boldsymbol{\delta}_i\}$.

We finally remark that there was an *independent* work (Tramèr & Boneh, 2019) which also proposed the formulation (10) for AT under multiple perturbations. However, what we propose here is the regularized formulation of (11). As will be evident later, the domain weights $\mathbf{w}$ in our formulation have strong interpretability, which learns the importance level of different attacks. Most significantly, our work has different motivation from (Tramèr & Boneh, 2019), and our idea applies to not only AT but also attack generation in Sec. 2.2.

## 3 Proposed Algorithm and Theory

In this section, we delve into technical details on how to efficiently solve problems of robust adversarial attacks given by the generic form (4) and problem (11) for generalized AT under mixed types of adversarial attacks.

### 3.1 ALTERNATING ONE-STEP PGD FOR ROBUST ADVERSARIAL ATTACK GENERATION

We propose the **a**lternating one-step **p**rojected **g**radient **d**escent (APGD) method (Algorithm 1) to solve problem (4). For clarity, we repeat problem (4) under the adversarial perturbation $\boldsymbol{\delta}$ and its constraint set $\mathcal{X}$ defined in Sec. 2.2,

$$\operatorname*{minimize}_{\boldsymbol{\delta} \in \mathcal{X}} \operatorname*{maximize}_{\mathbf{w} \in \mathcal{P}} \quad \sum_{i=1}^{K} w_i F_i(\boldsymbol{\delta}). \quad (12)$$

We show that at each iteration, APGD takes only one-step PGD for outer minimization and one-step projected gradient ascent for inner max-

---

**Algorithm 1** APGD to solve problem (4)
___
1: Input: given $\mathbf{w}^{(0)}$ and $\boldsymbol{\delta}^{(0)}$.
2: **for** $t = 1, 2, \ldots, T$ **do**
3:    *outer min.*: fixing $\mathbf{w} = \mathbf{w}^{(t-1)}$, call PGD (13) to update $\boldsymbol{\delta}^{(t)}$
4:    *inner max.*: fixing $\boldsymbol{\delta} = \boldsymbol{\delta}^{(t)}$, update $\mathbf{w}^{(t)}$ via (14)
5: **end for**

---

imization (namely, PGD for its negative objective function). We also show that each alternating step has a closed-form expression, and the main computational complexity stems from computing the gradient of the attack loss w.r.t. the input. Therefore, APGD is computationally efficient like PGD, which is commonly used for design of conventional single $\ell_p$-norm based adversarial attacks (Madry et al., 2017).

**Outer minimization** Considering $\mathbf{w} = \mathbf{w}^{(t-1)}$ and $F(\boldsymbol{\delta}) := \sum_{i=1}^{K} w_i^{(t-1)} F_i(\boldsymbol{\delta})$ in (4), we perform one-step PGD to update $\boldsymbol{\delta}$ at iteration $t$,

$$\boldsymbol{\delta}^{(t)} = \operatorname{proj}_{\mathcal{X}} \left( \boldsymbol{\delta}^{(t-1)} - \alpha \nabla_{\boldsymbol{\delta}} F(\boldsymbol{\delta}^{(t-1)}) \right), \quad (13)$$

where $\operatorname{proj}(\cdot)$ denotes the Euclidean projection operator, i.e., $\operatorname{proj}_{\mathcal{X}}(\mathbf{a}) = \arg\min_{\mathbf{x} \in \mathcal{X}} \|\mathbf{x} - \mathbf{a}\|_2^2$ at the point $\mathbf{a}$, $\alpha > 0$ is a given learning rate, and $\nabla_{\boldsymbol{\delta}}$ denotes the first order gradient w.r.t. $\boldsymbol{\delta}$.

In (13), the projection operation becomes the key to obtain the closed-form of the updating rule (13). Recall from Sec. 2.2 that $\mathcal{X} = \{\boldsymbol{\delta} | \|\boldsymbol{\delta}\|_p \leq \epsilon, \check{\mathbf{c}} \leq \boldsymbol{\delta} \leq \hat{\mathbf{c}}\}$, where $p \in \{0, 1, 2, \infty\}$, and $\check{\mathbf{c}} = -\mathbf{x}_0$ and $\hat{\mathbf{c}} = \mathbf{1} - \mathbf{x}_0$ (implying $\check{\mathbf{c}} \leq \mathbf{0} \leq \hat{\mathbf{c}}$). If $p = \infty$, then the projection function becomes the clip function. However, when $p \in \{0, 1, 2\}$, the closed-form of projection operation becomes non-trivial. In Proposition 1, we derive the solution of $\operatorname{proj}_{\mathcal{X}}(\mathbf{a})$ under different $\ell_p$ norms.

**Proposition 1.** *Given a point $\mathbf{a} \in \mathbb{R}^d$ and a constraint set $\mathcal{X} = \{\boldsymbol{\delta} | \|\boldsymbol{\delta}\|_p \leq \epsilon, \check{\mathbf{c}} \leq \boldsymbol{\delta} \leq \hat{\mathbf{c}}\}$, the Euclidean projection $\boldsymbol{\delta}^* = \operatorname{proj}_{\mathcal{X}}(\mathbf{a})$ has a closed-form solution when $p \in \{0, 1, 2\}$.*

**Proof**: See Appendix B. $\qquad\qquad\square$

**Inner maximization** By fixing $\boldsymbol{\delta} = \boldsymbol{\delta}^{(t)}$ and letting $\psi(\mathbf{w}) := \sum_{i=1}^{K} w_i F_i(\boldsymbol{\delta}^{(t)}) - \frac{\gamma}{2} \|\mathbf{w} - 1/K\|_2^2$ in problem (4), we then perform one-step PGD (w.r.t. $-\psi$) to update $\mathbf{w}$,

$$\mathbf{w}^{(t)} = \operatorname{proj}_{\mathcal{P}} \Big( \underbrace{\mathbf{w}^{(t-1)} + \beta \nabla_{\mathbf{w}} \psi(\mathbf{w}^{(t-1)})}_{\mathbf{b}} \Big) = (\mathbf{b} - \mu\mathbf{1})_+, \quad (14)$$

where $\beta > 0$ is a given learning rate, $\nabla_{\mathbf{w}} \psi(\mathbf{w}) = \boldsymbol{\phi}^{(t)} - \gamma(\mathbf{w} - 1/K)$, and $\boldsymbol{\phi}^{(t)} := [F_1(\boldsymbol{\delta}^{(t)}), \ldots, F_K(\boldsymbol{\delta}^{(t)})]^T$. In (14), the second equality holds due to the closed-form of projection operation onto the probabilistic simplex $\mathcal{P}$ (Parikh et al., 2014), where $(\cdot)_+$ denotes the elementwise non-negative operator, i.e., $(x)_+ = \max\{0, x\}$, and $\mu$ is the root of the equation $\mathbf{1}^T(\mathbf{b} - \mu\mathbf{1})_+ = 1$. Since $\mathbf{1}^T(\mathbf{b} - \min_i\{b_i\}\mathbf{1} + 1/K)_+ \geq \mathbf{1}^T\mathbf{1}/K = 1$, and $\mathbf{1}^T(\mathbf{b} - \max_i\{b_i\}\mathbf{1} + 1/K)_+ \leq \mathbf{1}^T\mathbf{1}/K = 1$, the root $\mu$ exists within the interval $[\min_i\{b_i\} - 1/K, \max_i\{b_i\} - 1/K]$ and can be found via the bisection method (Boyd & Vandenberghe, 2004).

**Convergence analysis** We remark that APGD follows the gradient primal-dual optimization framework (Lu et al., 2019a), and thus enjoys the same optimization guarantees. In Theorem 1, we demonstrate the convergence rate of Algorithm 1 for solving problem (4).

**Theorem 1.** *(inherited from primal-dual min-max optimization) Suppose that in problem (4) $F_i(\boldsymbol{\delta})$ has L-Lipschitz continuous gradients, and $\mathcal{X}$ is a convex compact set. Given learning rates $\alpha \leq \frac{1}{L}$ and $\beta < \frac{1}{\gamma}$, then the sequence $\{\boldsymbol{\delta}^{(t)}, \mathbf{w}^{(t)}\}_{t=1}^{T}$ generated by Algorithm 1 converges to a first-order stationary point[2] of problem (4) under the convergence rate $O(\frac{1}{T})$.*

---

[2]The stationarity is measured by the $\ell_2$ norm of gradient of the objective in (4) w.r.t. $(\boldsymbol{\delta}, \mathbf{w})$.

**Proof**: Note that the objective function of problem (4) is strongly concave w.r.t. $\mathbf{w}$ with parameter $\gamma$, and has $\gamma$-Lipschitz continuous gradients. Moreover, we have $\|\mathbf{w}\|_2 \leq 1$ due to $\mathbf{w} \in \mathcal{P}$. Using these facts and (Lu et al., 2019a, Theorem 1) or (Lu et al., 2019b, Theorem 1) completes the proof. $\qquad\square$

## 3.2 ALTERNATING MULTI-STEP PGD FOR GENERALIZED AT

We next propose the **a**lternating **m**ulti-step **p**rojected **g**radient **d**escent (AMPGD) method to solve the regularized version of problem (11), which is repeated as follows

$$\underset{\boldsymbol{\theta}}{\text{minimize}} \; \mathbb{E}_{(\mathbf{x},\mathbf{y})\in\mathcal{D}} \; \underset{\mathbf{w}\in\mathcal{P},\{\boldsymbol{\delta}_i\in\mathcal{X}_i\}}{\text{maximize}} \; \psi(\boldsymbol{\theta},\mathbf{w},\{\boldsymbol{\delta}_i\}) := \sum_{i=1}^{K} w_i f_{\text{tr}}(\boldsymbol{\theta},\boldsymbol{\delta}_i;\mathbf{x},y) - \frac{\gamma}{2}\|\mathbf{w}-\mathbf{1}/K\|_2^2. \quad (15)$$

Problem (15) is in a more general non-convex non-concave min-max setting, where the inner maximization involves both domain weights $\mathbf{w}$ and adversarial perturbations $\{\boldsymbol{\delta}_i\}$. It was shown in (Nouiehed et al., 2019) that the multi-step PGD is required for inner maximization in order to approximate the near-optimal solution. This is also in the similar spirit of AT (Madry et al., 2017), which executed multi-step PGD attack during inner maximization. We summarize AMPGD in Algorithm 2. At step 4 of Algorithm 2, each PGD step to update $\mathbf{w}$ and $\boldsymbol{\delta}$ can be decomposed as

---

**Algorithm 2** AMPGD to solve problem (15)

1: **Input**: given $\boldsymbol{\theta}^{(0)}$, $\mathbf{w}^{(0)}$, $\boldsymbol{\delta}^{(0)}$ and $K > 0$.
2: **for** $t = 1, 2, \ldots, T$ **do**
3:      given $\mathbf{w}^{(t-1)}$ and $\boldsymbol{\delta}^{(t-1)}$, perform SGD to update $\boldsymbol{\theta}^{(t)}$ (Madry et al., 2017)
4:      given $\boldsymbol{\theta}^{(t)}$, perform $R$-step PGD to update $\mathbf{w}^{(t)}$ and $\boldsymbol{\delta}^{(t)}$
5: **end for**

---

$$\mathbf{w}_r^{(t)} = \text{proj}_{\mathcal{P}}\left(\mathbf{w}_{r-1}^{(t)} + \beta\nabla_{\mathbf{w}}\psi(\boldsymbol{\theta}^{(t)},\mathbf{w}_{r-1}^{(t)},\{\boldsymbol{\delta}_{i,r-1}^{(t)}\})\right), \forall r \in [R],$$

$$\boldsymbol{\delta}_{i,r}^{(t)} = \text{proj}_{\mathcal{X}_i}\left(\boldsymbol{\delta}_{i,r-1}^{(t)} + \beta\nabla_{\boldsymbol{\delta}}\psi(\boldsymbol{\theta}^{(t)},\mathbf{w}_{r-1}^{(t)},\{\boldsymbol{\delta}_{i,r-1}^{(t)}\})\right), \forall r \in [R], \forall i \in [K]$$

where let $\mathbf{w}_1^{(t)} := \mathbf{w}^{(t-1)}$ and $\boldsymbol{\delta}_{i,1}^{(t)} := \boldsymbol{\delta}_i^{(t-1)}$. Here the subscript $t$ represents the iteration index of AMPGD, and the subscript $r$ denotes the iteration index of $R$-step PGD. Clearly, the above projection operations can be derived for closed-form expressions through (14) and Lemma 1. To the best of our knowledge, it is still an open question to build theoretical convergence guarantees for solving the general non-convex non-concave min-max problem like (15), except the work (Nouiehed et al., 2019) which proposed $O(1/T)$ convergence rate if the objective function satisfies Polyak-Łojasiewicz conditions (Karimi et al., 2016).

**Improved robustness via diversified $\ell_p$ attacks.** It was recently shown in (Kariyappa & Qureshi, 2019; Pang et al., 2019) that the diversity of individual neural networks improves adversarial robustness of an ensemble model. Spurred by that, one may wonder if the promotion of diversity among $\ell_p$ attacks is beneficial to adversarial robustness? We measure the diversity between adversarial attacks through the similarity between perturbation directions, namely, input gradients $\{\nabla_{\boldsymbol{\delta}_i} f_{\text{tr}}(\boldsymbol{\theta},\boldsymbol{\delta}_i;\mathbf{x},y)\}_i$ in (15). We find that there exists a strong correlation between input gradients for different $\ell_p$ attacks. Thus, we propose to enhance their diversity through the orthogonality-promoting regularizer used for encouraging diversified prediction of ensemble models in (Pang et al., 2019),

$$h(\boldsymbol{\theta},\{\boldsymbol{\delta}_i\};\mathbf{x},y) := \log\det(\mathbf{G}^T\mathbf{G}), \quad (16)$$

where $\mathbf{G} \in \mathbb{R}^{d \times K}$ is a $d \times K$ matrix, each column of which corresponds to a normalized input gradient $\nabla_{\boldsymbol{\delta}_i} f_{\text{tr}}(\boldsymbol{\theta},\boldsymbol{\delta}_i;\mathbf{x},y)$ for $i \in [K]$, and $h(\boldsymbol{\theta},\{\boldsymbol{\delta}_i\};\mathbf{x},y)$ reaches the maximum value 0 as input gradients become orthogonal. With the aid of (16), we modify problem (15) to

$$\underset{\boldsymbol{\theta}}{\text{minimize}} \; \mathbb{E}_{(\mathbf{x},\mathbf{y})\in\mathcal{D}} \; \underset{\mathbf{w}\in\mathcal{P},\{\boldsymbol{\delta}_i\in\mathcal{X}_i\}}{\text{maximize}} \; \psi(\boldsymbol{\theta},\mathbf{w},\{\boldsymbol{\delta}_i\}) + \lambda h(\boldsymbol{\theta},\{\boldsymbol{\delta}_i\};\mathbf{x},y). \quad (17)$$

The rationale behind (17) is that the adversary aims to enhance the effectiveness of attacks from diversified perturbation directions (inner maximization), while the defender robustifies the model $\boldsymbol{\theta}$, which makes diversified attacks less effective (outer minimization).

## 4 EXPERIMENTS

In this section, we first evaluate the proposed min-max optimization strategy on three attack tasks. We show that our approach leads to substantial improvement compared with state-of-the-art attack

**Table 1:** Comparison of average and min-max (APGD) ensemble attack over four models on MNIST and CIFAR-10. Acc (%) represents the test accuracy of classifiers on adversarial examples. Here we set the iterations of APGD as 50 for attack generation. The learning rates $\alpha, \beta$ and regularization factor $\gamma$ are provided in Appendix C.2.

**(a) MNIST**

| Box constraint | Opt. | $Acc_A$ | $Acc_B$ | $Acc_C$ | $Acc_D$ | $ASR_{all}$ | Lift ($\uparrow$) |
|---|---|---|---|---|---|---|---|
| $\ell_0$ ($\epsilon = 30$) | avg. | 7.03 | 1.51 | 11.27 | 2.48 | 84.03 | - |
| | min max | 3.65 | 2.36 | 4.99 | 3.11 | **91.97** | **9.45%** |
| $\ell_1$ ($\epsilon = 20$) | avg. | 20.79 | 0.15 | 21.48 | 6.70 | 69.31 | - |
| | min max | 6.12 | 2.53 | 8.43 | 5.11 | **89.16** | **28.64%** |
| $\ell_2$ ($\epsilon = 3.0$) | avg. | 6.88 | 0.03 | 26.28 | 14.50 | 69.12 | - |
| | min max | 1.51 | 0.89 | 3.50 | 2.06 | **95.31** | **37.89%** |
| $\ell_\infty$ ($\epsilon = 0.2$) | avg. | 1.05 | 0.07 | 41.10 | 35.03 | 48.17 | - |
| | min max | 2.47 | 0.37 | 7.39 | 5.81 | **90.16** | **87.17%** |

**(b) CIFAR-10**

| Box constraint | Opt. | $Acc_A$ | $Acc_B$ | $Acc_C$ | $Acc_D$ | $ASR_{all}$ | Lift ($\uparrow$) |
|---|---|---|---|---|---|---|---|
| $\ell_0$ ($\epsilon = 50$) | avg. | 27.86 | 3.15 | 5.16 | 6.17 | 65.16 | - |
| | min max | 18.74 | 8.66 | 9.64 | 9.70 | **71.44** | **9.64%** |
| $\ell_1$ ($\epsilon = 30$) | avg. | 32.92 | 2.07 | 5.55 | 6.36 | 59.74 | - |
| | min max | 12.46 | 3.74 | 5.62 | 5.86 | **78.65** | **31.65%** |
| $\ell_2$ ($\epsilon = 2.0$) | avg. | 24.3 | 1.51 | 4.59 | 4.20 | 69.55 | - |
| | min max | 7.17 | 3.03 | 4.65 | 5.14 | **83.95** | **20.70%** |
| $\ell_\infty$ ($\epsilon = 0.05$) | avg. | 19.69 | 1.55 | 5.61 | 4.26 | 73.29 | - |
| | min max | 7.21 | 2.68 | 4.74 | 4.59 | **84.36** | **15.10%** |

methods such as ensemble PGD (Liu et al., 2018) and expectation over transformation (EOT) (Athalye et al., 2018b; Brown et al., 2017; Athalye et al., 2018a). We next demonstrate the effectiveness of the generalized AT for multiple types of adversarial perturbations. We show that the use of trainable domain weights in problem (15) can automatically adjust the risk level of different attacks during the training process even if the defender lacks prior knowledge on the strength of these attacks. We also show that the promotion of diversity of $\ell_p$ attacks help improve adversarial robustness further.

We thoroughly evaluate our APGD/AMPGD algorithm on MNIST and CIFAR-10. A set of diverse image classifiers (denoted from Model A to Model H) are trained, including multi-layer perceptrons (MLP), All-CNNs (Springenberg et al., 2015), LeNet (Lecun et al., 1998), LeNetV2, VGG16 (Simonyan & Zisserman, 2015), ResNet50 (He et al., 2016), Wide-ResNet (Madry et al., 2017; Zagoruyko & Komodakis, 2016) and GoogLeNet (Szegedy et al., 2015). More details about model architectures and training process are provided in Appendix C.1.

## 4.1 Robust adversarial attacks

Most current works play a min-max game from a defender's perspective, i.e., adversarial training. However, we show the great strength of min-max optimization also lies at the side of attack generation. Note that problem formulations (5)-(7) are applicable to both *untargeted* and *targeted* attack. Here we focus on the former setting and use C&W loss function (Carlini & Wagner, 2017; Madry et al., 2017). The details of crafting adversarial examples are available in Appendix C.2.

**Ensemble attack over multiple models** We craft adversarial examples against an ensemble of known classifiers. The work (Liu et al., 2018, 5th place at CAAD-18) proposed an ensemble PGD attack, which assumed equal importance among different models, namely, $w_i = 1/K$ in problem (1). Throughout this task, we measure the attack performance via **$ASR_{all}$** - the attack success rate (ASR) of fooling model ensembles simultaneously. Compared to the ensemble PGD attack (Liu et al., 2018), **our approach results in 40.79% and 17.48% $ASR_{all}$ improvement averaged over different $\ell_p$-norm constraints on MNIST and CIFAR-10**, respectively. In what follows, we provide more detailed experiment results and analysis.

In Table 1, we show that our min-max APGD significantly outperforms ensemble PGD in $ASR_{all}$. Taking $\ell_\infty$-attack on MNIST as an example, our min-max attack leads to a 90.16% $ASR_{all}$, which largely outperforms 48.17% (ensemble PGD). The reason is that Model C, D are more difficult to attack, which can be observed from their higher test accuracy on adversarial examples. As a result, although the adversarial examples crafted by assigning equal weights over multiple models are able to attack {A, B} well, they achieve a much lower ASR (i.e., 1 - Acc) in {C, D}. By contrast, APGD automatically handles the worst case {C, D} by slightly sacrificing the performance on {A, B}: 31.47% averaged ASR improvement on {C, D} versus 0.86% degradation on {A, B}. More results on CIFAR-10 and more complicated DNNs (e.g., GoogLeNet) are provided in Appendix D.

Figure 1 depicts the ASR of four models under average/min-max attacks as well as the distribution of domain weights during attack generation. For ensemble PGD (Figure 1a), Model C and D are attacked insufficiently, leading to relatively low ASR and thus weak ensemble performance. By contrast, APGD (Figure 1b) will encode the difficulty level to attack different models based on the current attack loss. It dynamically adjusts the weight $w_i$ as shown in Figure 1c. For instance, the weight for Model D is first raised to $0.45$ because D is difficult to attack initially. Then it decreases to

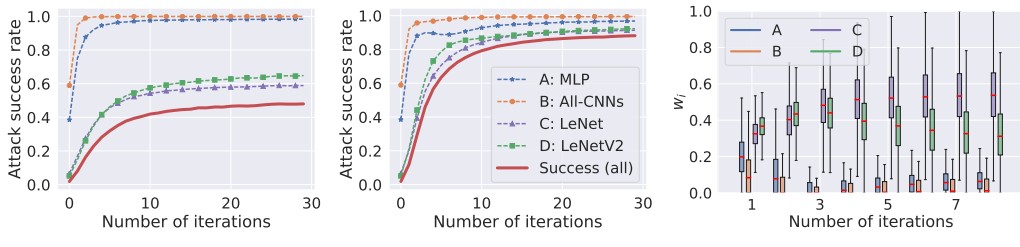

**(a)** average case        **(b)** min max        **(c)** weight $\{w_i\}$

**Figure 1:** Ensemble attack against four DNN models on MNIST. (a) & (b): Attack success rate of adversarial examples generated by average (ensemble PGD) or min-max (APGD) attack method. (c): Boxplot of weight $w$ in APGD adversarial loss for four models. Here we adopt the same $\ell_\infty$-attack as Table 1.

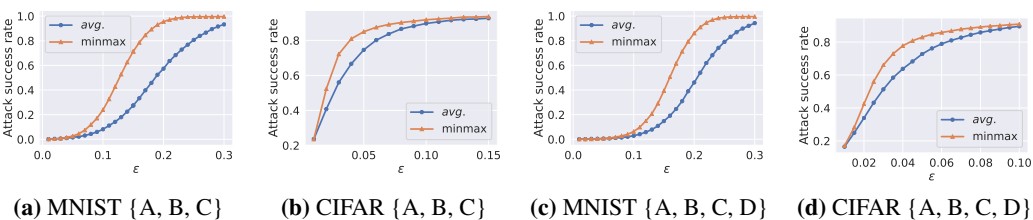

**(a)** MNIST {A, B, C}    **(b)** CIFAR {A, B, C}    **(c)** MNIST {A, B, C, D}    **(d)** CIFAR {A, B, C, D}

**Figure 2:** ASR of average and min-max $\ell_\infty$ ensemble attack versus maximum perturbation magnitude $\epsilon$.

0.3 once Model D encounters the sufficient attack power and the corresponding attack performance is no longer improved. It is worth noticing that APGD is highly efficient because $w_i$ converges after a small number of iterations. To perform a boarder evaluation, we repeat the above experiments ($\ell_\infty$ norm) under different $\epsilon$ in Figure 2. The ASR of min-max strategy is consistently better or on part with the average strategy. Moreover, APGD achieves more significant improvement when moderate $\epsilon$ is chosen: MNIST ($\epsilon \in [0.15, 0.25]$) and CIFAR-10 ($\epsilon \in [0.03, 0.05]$).

Lastly, we highlight that tracking domain weights $\mathbf{w}$ provides us novel insights for model robustness and understanding attack procedure. From our theory, a model with higher robustness always corresponds to a larger $w$ because its loss is hard to attack and becomes the "worst" term. This hypothesis can be verified empirically. According to Figure 1c, we have $w_c > w_d > w_a > w_b$ – indicating a decrease in model robustness for C, D, A and B, which is exactly verified by $\text{Acc}_C > \text{Acc}_D > \text{Acc}_A > \text{Acc}_B$ in Table 1 ($\ell_\infty$-norm).

**Universal perturbation over multiple examples**    We evaluate APGD in universal perturbation on MNIST and CIFAR-10, where 10,000 test images are randomly divided into equal-size groups (containing $K$ images per group) for universal perturbation. We measure two types of ASR (%), **ASR**$_{avg}$ and **ASR**$_{gp}$. Here the former represents the ASR averaged over all images in all groups, and the latter signifies the ASR averaged over all groups but a successful attack is counted under a more restricted condition: images within each group must be successfully attacked simultaneously by universal perturbation. **When $K = 5$, our approach achieves 42.63% and 35.21% improvement over the averaging strategy under MNIST and CIFAR-10, respectively.**

In Table 2, we compare the proposed min-max strategy (APGD) with the averaging strategy on the attack performance of generated universal perturbations. As we can see, our method always achieves higher $\text{ASR}_{gp}$ for different values of $K$. The universal perturbation generated from APGD can successfully attack 'hard' images (on which the average-based PGD attack fails) by self-adjusting domain weights, and thus leads to a higher $\text{ASR}_{gp}$. Besides, the min-max universal perturbation also offers interpretability of "image robustness" by associating domain weights with image visualization; see Figure A9 and A10 (Appendix F) for an example in which the large domain weight corresponds to the MNIST letter with clear appearance (e.g., bold letter).

**Robust adversarial attack over data transformations**    EOT (Athalye et al., 2018a) achieves state-of-the-art performance in producing adversarial examples robust to data transformations. From (7), we could derive EOT as a special case when the weights satisfy $w_i = 1/K$ (average case). For each input sample (*ori*), we transform the image under a series of functions, e.g., flipping horizontally

**Table 2:** Comparison of average and minmax optimization on universal perturbation over multiple input examples. The adversarial examples are generated by 20-step $\ell_\infty$-APGD with $\alpha = \frac{1}{6}, \beta = \frac{1}{50}$ and $\gamma = 4$.

| Setting | | | $K=2$ | | | $K=4$ | | | $K=5$ | | | $K=10$ | | |
|---|---|---|---|---|---|---|---|---|---|---|---|---|---|---|
| Dataset | Model | Opt. | $\text{ASR}_{avg}$ | $\text{ASR}_{gp}$ | Lift ($\uparrow$) | $\text{ASR}_{avg}$ | $\text{ASR}_{gp}$ | Lift ($\uparrow$) | $\text{ASR}_{avg}$ | $\text{ASR}_{gp}$ | Lift ($\uparrow$) | $\text{ASR}_{avg}$ | $\text{ASR}_{gp}$ | Lift ($\uparrow$) |
| MNIST | MLP | avg. | 97.19 | 94.48 | - | 85.13 | 56.64 | - | 79.11 | 38.05 | - | 60.53 | 3.50 | - |
| | | min max | 98.15 | **96.96** | **2.62%** | 83.76 | **72.32** | **27.68%** | 72.28 | **53.70** | **41.13%** | 30.10 | **6.70** | **91.43%** |
| | All-CNNs | avg. | 97.76 | 95.52 | - | 85.19 | 51.92 | - | 80.02 | 31.25 | - | 65.79 | 2.10 | - |
| | | min max | 99.69 | **99.38** | **4.04%** | 90.11 | **75.64** | **45.69%** | 80.21 | **53.50** | **71.20%** | 43.54 | **4.30** | **104.8%** |
| | LeNet | avg. | 94.78 | 89.96 | - | 62.12 | 28.72 | - | 51.84 | 19.15 | - | 30.29 | 4.30 | - |
| | | min max | 96.60 | **94.58** | **5.14%** | 55.50 | **36.72** | **27.86%** | 42.79 | **25.80** | **34.73%** | 22.48 | **7.20** | **67.44%** |
| | LeNetV2 | avg. | 94.72 | 90.04 | - | 61.59 | 26.60 | - | 50.42 | 17.05 | - | 26.49 | 4.80 | - |
| | | min max | 97.33 | **95.68** | **6.26%** | 55.38 | **35.52** | **33.53%** | 40.22 | **21.05** | **23.46%** | 19.73 | **7.10** | **47.92%** |
| CIFAR-10 | All-CNNs | avg. | 91.09 | 83.08 | - | 85.66 | 54.72 | - | 82.76 | 40.20 | - | 71.22 | 4.50 | - |
| | | min max | 92.22 | **85.98** | **3.49%** | 87.63 | **65.80** | **20.25%** | 85.02 | **55.74** | **38.66%** | 65.64 | **11.80** | **162.2%** |
| | LeNetV2 | avg. | 93.26 | 86.90 | - | 90.04 | 66.12 | - | 88.28 | 55.00 | - | 72.02 | 8.90 | - |
| | | min max | 93.34 | **87.08** | **0.21%** | 91.91 | **71.64** | **8.35%** | 91 | **63.55** | **15.55%** | 82.85 | **25.10** | **182.0%** |
| | VGG16 | avg. | 90.76 | 82.56 | - | 89.36 | 63.92 | - | 88.74 | 55.20 | - | 85.86 | 22.40 | - |
| | | min max | 92.40 | **85.92** | **4.07%** | 90.04 | **70.40** | **10.14%** | 88.97 | **63.30** | **14.67%** | 79.07 | **30.80** | **37.50%** |
| | GoogLeNet | avg. | 85.02 | 72.48 | - | 75.20 | 32.68 | - | 71.82 | 19.60 | - | 59.01 | 0.40 | - |
| | | min max | 87.08 | **77.82** | **7.37%** | 77.05 | **46.20** | **41.37%** | 71.20 | **33.70** | **71.94%** | 45.46 | **2.40** | **600.0%** |

**Table 3:** Comparison of average and min-max optimization on robust attack over multiple data transformations on CIFAR-10. Acc (%) represents the test accuracy of classifiers on adversarial examples (20-step $\ell_\infty$-APGD ($\epsilon = 0.03$) with $\alpha = \frac{1}{2}, \beta = \frac{1}{100}$ and $\gamma = 10$) under different transformations.

| Model | Opt. | $\text{Acc}_{ori}$ | $\text{Acc}_{flh}$ | $\text{Acc}_{flv}$ | $\text{Acc}_{bri}$ | $\text{Acc}_{gam}$ | $\text{Acc}_{crop}$ | $\text{ASR}_{avg}$ | $\text{ASR}_{gp}$ | Lift ($\uparrow$) |
|---|---|---|---|---|---|---|---|---|---|---|
| A | avg. | 10.80 | 21.93 | 14.75 | 11.52 | 10.66 | 20.03 | 85.05 | 55.88 | - |
| | min max | 12.14 | 18.05 | 13.61 | 13.52 | 11.99 | 16.78 | 85.65 | **60.03** | **7.43%** |
| B | avg. | 5.49 | 11.56 | 9.51 | 5.43 | 5.75 | 15.89 | 91.06 | 72.21 | - |
| | min max | 6.22 | 8.61 | 9.74 | 6.35 | 6.42 | 11.99 | 91.78 | **77.43** | **7.23%** |
| C | avg. | 7.66 | 21.88 | 15.50 | 8.15 | 7.87 | 15.36 | 87.26 | 56.51 | - |
| | min max | 8.51 | 14.75 | 13.88 | 9.16 | 8.58 | 13.35 | 88.63 | **63.58** | **12.51%** |
| D | avg. | 8.00 | 20.47 | 13.46 | 7.73 | 8.52 | 15.90 | 87.65 | 61.13 | - |
| | min max | 9.19 | 13.18 | 12.72 | 8.79 | 9.18 | 13.11 | 88.97 | **67.49** | **10.40%** |

(*flh*) or vertically (*flv*), adjusting brightness (*bri*), performing gamma correction (*gam*) and cropping (*crop*), and group each image with its transformed variants. Similar to universal perturbation, $\text{ASR}_{avg}$ and $\text{ASR}_{gp}$ are reported to measure the ASR over all transformed images and groups of transformed images (each group is successfully attacked signifies successfully attacking an example under all transformers). In Table 3, **compared to EOT, our approach leads to 9.39% averaged lift in $\text{ASR}_{gp}$** over given models on CIFAR-10 by optimizing the weights for various transformations. Due to limited space, we leave the details of transformers in Append C.3 and the results under randomness (e.g., flipping images randomly *w.p.* 0.8; randomly clipping the images at specific range) in Appendix D.

## 4.2 ADVERSARIAL TRAINING FOR MULTIPLE ADVERSARIAL PERTURBATIONS

Compared to vanilla AT, we show the generalized AT scheme produces models robust to multiple types of perturbation, thus leads to stronger "overall robustness". We measure the training performance using two types of Acc (%): $\textbf{Acc}^{\text{max}}_{\text{adv}}$ and $\textbf{Acc}^{\text{avg}}_{\text{adv}}$, where $\text{Acc}^{\text{max}}_{\text{adv}}$ denotes the test accuracy over examples with the strongest perturbation ($\ell_\infty$ or $\ell_2$), and $\text{Acc}^{\text{avg}}_{\text{adv}}$ denotes the averaged test accuracy over examples with all types of perturbations ($\ell_\infty$ and $\ell_2$). Moreover, we measure the **overall worst-case robustness** $S_\epsilon$ in terms of the area under the curve '$\text{Acc}^{\text{max}}_{\text{adv}}$ vs. $\epsilon$' (see Figure 3b).

In Table 4, we present the test accuracy of MLP in different training schemes: a) natural training, b) single-norm: vanilla AT ($\ell_\infty$ or $\ell_2$), c) multi-norm: generalized AT ($avg$ and $\min\max$), and d) generalized AT with diversity-promoting attack regularization (DPAR, $\lambda = 0.1$ in problem (16)). If the adversary only performs single-type attack, training and testing on the same attack type leads to the best performance (diagonal of $\ell_\infty$-$\ell_2$ block). However, when facing $\ell_\infty$ and $\ell_2$ attacks simultaneously, multi-norm generalized AT achieves better $\text{Acc}^{\text{max}}_{\text{adv}}$ and $\text{Acc}^{\text{avg}}_{\text{adv}}$ than single-norm AT. In particular, the min-max strategy slightly outperforms the averaging strategy under multiple perturbation norms.

**Table 4:** Adversarial training of MNIST models on single attacks ($\ell_\infty$ and $\ell_2$) and multiple attacks ($avg.$ and min max). The perturbation magnitude $\epsilon$ for $\ell_\infty$ and $\ell_2$ attacks are 0.2 and 2.0, respectively. Top 2 test accuracy on each metric are highlighted. Complete table for varied $\epsilon$ is given in Table A8 (Appendix E).

**(a) MLP**

| Opt. | Acc. | Acc-$\ell_\infty$ | Acc-$\ell_2$ | Acc$_{adv}^{max}$ | Acc$_{adv}^{avg}$ |
|---|---|---|---|---|---|
| natural | 98.30 | 2.70 | 13.86 | 0.85 | 8.28 |
| $\ell_\infty$ | 98.08 | **77.70** | 69.17 | 66.34 | 73.43 |
| $\ell_2$ | 98.72 | 70.03 | **81.74** | 69.14 | 75.88 |
| $avg.$ | 98.62 | 75.09 | 79.00 | 72.23 | 77.05 |
| + DPAR | 98.50 | 76.75 | 79.67 | **74.14** | **78.21** |
| min max | 98.59 | 75.96 | 79.15 | 73.43 | 77.55 |
| + DPAR | 98.58 | **76.92** | **79.74** | **74.29** | **78.35** |

**(b) LeNet**

| Opt. | Acc. | Acc-$\ell_\infty$ | Acc-$\ell_2$ | Acc$_{adv}^{max}$ | Acc$_{adv}^{avg}$ |
|---|---|---|---|---|---|
| natural | 99.25 | 17.93 | 39.32 | 17.57 | 28.63 |
| $\ell_\infty$ | 99.18 | **93.80** | 78.97 | 78.80 | 86.39 |
| $\ell_2$ | 99.22 | 85.84 | **87.31** | 84.06 | 86.58 |
| $avg.$ | 99.22 | 88.96 | 85.59 | 84.29 | 87.28 |
| + DPAR | 99.25 | 89.96 | **86.49** | **85.44** | **88.23** |
| min max | 99.32 | 89.21 | 85.98 | 84.82 | 87.60 |
| + DPAR | 99.22 | **90.19** | 86.47 | **85.47** | **88.33** |

**(a) weight $\{w_i\}$**  **(b) Acc$_{adv}^{max}$**  **(c) Acc$_{adv}^{avg}$**

**Figure 3:** (a): Violin plot of weight $w$ in APGD versus perturbation magnitude $\epsilon$ of $\ell_2$-attack in AT; (b) & (c): Robustness of MLP under different AT schemes. Supplementary result for LeNet is provided in Figure A2 (Appendix E).

DPAR further boosts the adversarial test accuracy, which implies that the promotion of diversified $\ell_p$ attacks is a beneficial supplement to adversarial training.

In Figure 3, we offer deeper insights on the performance of generalized AT. During the training procedure we fix $\epsilon_{\ell_\infty}$ ($\epsilon$ for $\ell_\infty$ attack during training) as 0.2, and change $\epsilon_{\ell_2}$ from 0.2 to 5.6 ($\epsilon_{\ell_\infty} \times \sqrt{d}$) so that the $\ell_\infty$ and $\ell_2$ balls are not completely overlapped (Araujo et al., 2019). In Figure 3a, as $\epsilon_{\ell_2}$ increases, $\ell_2$-attack becomes stronger so the corresponding $w$ also increases, which is consistent with min-max spirit – defending the strongest attack. We remark that min max or $avg$ training does not always lead to the best performance on Acc$_{adv}^{max}$ and Acc$_{adv}^{avg}$, especially when the strengths of two attacks diverge greatly (see Table A8). This can be explained by the large overlapping between $\ell_\infty$ and $\ell_2$ balls (see Figure A3). However, Figure 3b and 3c show that AMPGD is able to achieve a rather robust model no matter how $\epsilon$ changes (red lines), which empirically verifies the effectiveness of our proposed training scheme. In terms of the area-under-the-curve measure $S_\epsilon$, **AMPGD achieves the highest worst-case robustness: 6.27% and 17.64% improvement compared to the vanilla AT with $\ell_\infty$ and $\ell_2$ attacks**. Furthermore, we show in Figure A4a that our min-max scheme leads to faster convergence than the averaging scheme due to the benefit of self-adjusted domain weights.

## 5 CONCLUSION

In this paper, we propose a general min-max framework applicable to both adversarial attack and defense settings. We show that many problem setups can be re-formulated under this general framework. Extensive experiments show that proposed algorithms lead to significant improvement on multiple attack and defense tasks compared with previous state-of-the-art approaches. In particular, we obtain 17.48%, 35.21% and 9.39% improvement on attacking model ensembles, devising universal perturbation to input samples, and data transformations under CIFAR-10, respectively. Our min-max scheme also generalizes adversarial training (AT) for multiple types of adversarial attacks, attaining faster convergence and better robustness compared to the vanilla AT and the average strategy. Moreover, our approach provides a holistic tool for self-risk assessment by learning domain weights.

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

## CONTENTS

## LIST OF FIGURES

## LIST OF TABLES

# A    PROOF OF LEMMA 1

**Lemma 1.** *Problem* (10) *is equivalent to*

$$\underset{\boldsymbol{\theta}}{\text{minimize}} \; \mathbb{E}_{(\mathbf{x},\mathbf{y})\in\mathcal{D}} \; \underset{\mathbf{w}\in\mathcal{P}, \{\boldsymbol{\delta}_i\in\mathcal{X}_i\}}{\text{maximize}} \; \sum_{i=1}^{K} w_i f_{\text{tr}}(\boldsymbol{\theta}, \boldsymbol{\delta}_i; \mathbf{x}, y),$$

*where* $\mathbf{w} \in \mathbb{R}^K$ *represent domain weights, and* $\mathcal{P}$ *has been defined in* (1).

***Proof of Lemma 1:***

Similar to (1), problem (10) is equivalent to

$$\underset{\boldsymbol{\theta}}{\text{minimize}} \; \mathbb{E}_{(\mathbf{x},\mathbf{y})\in\mathcal{D}} \; \underset{\mathbf{w}\in\mathcal{P}}{\text{maximize}} \sum_{i=1}^{K} w_i F_i(\boldsymbol{\theta}). \tag{18}$$

Recall that $F_i(\boldsymbol{\theta}) := \text{maximize}_{\boldsymbol{\delta}_i\in\mathcal{X}_i} \; f_{\text{tr}}(\boldsymbol{\theta}, \boldsymbol{\delta}_i; \mathbf{x}, y)$, problem can then be written as

$$\underset{\boldsymbol{\theta}}{\text{minimize}} \; \mathbb{E}_{(\mathbf{x},\mathbf{y})\in\mathcal{D}} \; \underset{\mathbf{w}\in\mathcal{P}}{\text{maximize}} \sum_{i=1}^{K} [w_i \underset{\boldsymbol{\delta}_i\in\mathcal{X}_i}{\text{maximize}} \; f_{\text{tr}}(\boldsymbol{\theta}, \boldsymbol{\delta}_i; \mathbf{x}, y)]. \tag{19}$$

According to proof by contradiction, it is clear that problem (19) is equivalent to

$$\underset{\boldsymbol{\theta}}{\text{minimize}} \; \mathbb{E}_{(\mathbf{x},\mathbf{y})\in\mathcal{D}} \; \underset{\mathbf{w}\in\mathcal{P}, \{\boldsymbol{\delta}_i\in\mathcal{X}_i\}}{\text{maximize}} \; \sum_{i=1}^{K} w_i f_{\text{tr}}(\boldsymbol{\theta}, \boldsymbol{\delta}_i; \mathbf{x}, y). \tag{20}$$

$\square$

# B    PROOF OF PROPOSITION 1

**Proposition 1.** *Given a point* $\mathbf{a} \in \mathbb{R}^d$ *and a constraint set* $\mathcal{X} = \{\boldsymbol{\delta}|\|\boldsymbol{\delta}\|_p \le \epsilon, \check{\mathbf{c}} \le \boldsymbol{\delta} \le \hat{\mathbf{c}}\}$, *the Euclidean projection* $\boldsymbol{\delta}^* = \text{proj}_{\mathcal{X}}(\mathbf{a})$ *has the closed-form solution when* $p \in \{0, 1, 2\}$.

*1) If* $p = 1$, *then* $\boldsymbol{\delta}^*$ *is given by*

$$\delta_i^* = \begin{cases} P_{[\check{c}_i, \hat{c}_i]}(a_i) & \sum_{i=1}^d |P_{[\check{c}_i, \hat{c}_i]}(a_i)| \le \epsilon \\ P_{[\check{c}_i, \hat{c}_i]}(\text{sign}(a_i) \max\{|a_i| - \lambda_1, 0\}) & \text{otherwise}, \end{cases} \tag{21}$$

*where* $\mathbf{x}_i$ *denotes the ith element of a vector* $\mathbf{x}$; $P_{[\check{c}_i, \hat{c}_i]}(\cdot)$ *denotes the clip function over the interval* $[\check{c}_i, \hat{c}_i]$; $\text{sign}(x) = 1$ *if* $x \ge 0$, *otherwise* $0$; $\lambda_1 \in (0, \max_i |a_i| - \epsilon/d]$ *is the root of* $\sum_{i=1}^d |P_{[\check{c}_i, \hat{c}_i]}(\text{sign}(a_i) \max\{|a_i| - \lambda_1, 0\})| = \epsilon$.

*2) If* $p = 2$, *then* $\boldsymbol{\delta}^*$ *is given by*

$$\delta_i^* = \begin{cases} P_{[\check{c}_i, \hat{c}_i]}(a_i) & \sum_{i=1}^d (P_{[\check{c}_i, \hat{c}_i]}(a_i))^2 \le \epsilon^2 \\ P_{[\check{c}_i, \hat{c}_i]}(a_i/(\lambda_2 + 1)) & \text{otherwise}, \end{cases} \tag{22}$$

*where* $\lambda_2 \in (0, \|\mathbf{a}\|_2/\epsilon - 1]$ *is the root of* $\sum_{i=1}^d (P_{[\check{c}_i, \hat{c}_i]}(a_i/(\lambda_2 + 1)))^2 = \epsilon^2$.

*3) If* $p = 0$ *and* $\epsilon \in \mathbb{N}_+$, *then* $\boldsymbol{\delta}^*$ *is given by*

$$\delta_i^* = \begin{cases} \delta_i' & \eta_i \ge [\boldsymbol{\eta}]_\epsilon \\ 0 & \text{otherwise}, \end{cases} \qquad \eta_i = \begin{cases} \sqrt{2a_i\check{c}_i - \check{c}_i^2} & a_i < \check{c}_i \\ \sqrt{2a_i\hat{c}_i - \hat{c}_i^2} & a_i > \hat{c}_i \\ |a_i| & \text{otherwise}. \end{cases} \tag{23}$$

*where* $[\boldsymbol{\eta}]_\epsilon$ *denotes the* $\epsilon$-*th largest element of* $\boldsymbol{\eta}$, *and* $\delta_i' = P_{[\check{c}_i, \hat{c}_i]}(a_i)$.

***Proof of Proposition 1:***

$\ell_1$ **norm** When we find the Euclidean projection of $\mathbf{a}$ onto the set $\mathcal{X}$, we solve

$$\underset{\boldsymbol{\delta}}{\text{minimize}} \quad \frac{1}{2}\|\boldsymbol{\delta} - \mathbf{a}\|_2^2 + I_{[\check{\mathbf{c}}, \hat{\mathbf{c}}]}(\boldsymbol{\delta}) \\ \text{subject to} \quad \|\boldsymbol{\delta}\|_1 \leq \epsilon, \tag{24}$$

where $I_{[\check{\mathbf{c}}, \hat{\mathbf{c}}]}(\cdot)$ is the indicator function of the set $[\check{\mathbf{c}}, \hat{\mathbf{c}}]$. The Langragian of this problem is

$$L = \frac{1}{2}\|\boldsymbol{\delta} - \mathbf{a}\|_2^2 + I_{[\check{\mathbf{c}}, \hat{\mathbf{c}}]}(\boldsymbol{\delta}) + \lambda_1(\|\boldsymbol{\delta}\|_1 - \epsilon) \tag{25}$$

$$= \sum_{i=1}^{d}(\frac{1}{2}(\delta_i - a_i)^2 + \lambda_1|\delta_i| + I_{[\check{c}_i, \hat{c}_i]}(\delta_i)) - \lambda_1\epsilon. \tag{26}$$

The minimizer $\boldsymbol{\delta}^*$ minimizes the Lagrangian, it is obtained by elementwise soft-thresholding

$$\delta_i^* = P_{[\check{c}_i, \hat{c}_i]}(\text{sign}(a_i)\max\{|a_i| - \lambda_1, 0\}).$$

where $\mathbf{x}_i$ is the $i$th element of a vector $\mathbf{x}$, $P_{[\check{c}_i, \hat{c}_i]}(\cdot)$ is the clip function over the interval $[\check{c}_i, \hat{c}_i]$.

The primal, dual feasibility and complementary slackness are

$$\lambda_1 = 0, \|\boldsymbol{\delta}\|_1 = \sum_{i=1}^{d}|\delta_i| = \sum_{i=1}^{d}|P_{[\check{c}_i, \hat{c}_i]}(a_i)| \leq \epsilon \tag{27}$$

$$\text{or } \lambda_1 > 0, \|\boldsymbol{\delta}\|_1 = \sum_{i=1}^{d}|\delta_i| = \sum_{i=1}^{d}|P_{[\check{c}_i, \hat{c}_i]}(\text{sign}(a_i)\max\{|a_i| - \lambda_1, 0\})| = \epsilon. \tag{28}$$

If $\sum_{i=1}^{d}|P_{[\check{c}_i, \hat{c}_i]}(a_i)| \leq \epsilon$, $\delta_i^* = P_{[\check{c}_i, \hat{c}_i]}(a_i)$. Otherwise $\delta_i^* = P_{[\check{c}_i, \hat{c}_i]}(\text{sign}(a_i)\max\{|a_i| - \lambda_1, 0\})$, where $\lambda_1$ is given by the root of the equation $\sum_{i=1}^{d}|P_{[\check{c}_i, \hat{c}_i]}(\text{sign}(a_i)\max\{|a_i| - \lambda_1, 0\})| = \epsilon$. Bisection method can be used to solve the above equation for $\lambda_1$, starting with the initial interval $(0, \max_i |a_i| - \epsilon/d]$. Since $\sum_{i=1}^{d}|P_{[\check{c}_i, \hat{c}_i]}(\text{sign}(a_i)\max\{|a_i| - 0, 0\})| = \sum_{i=1}^{d}|P_{[\check{c}_i, \hat{c}_i]}(a_i)| > \epsilon$ in this case, and $\sum_{i=1}^{d}|P_{[\check{c}_i, \hat{c}_i]}(\text{sign}(a_i)\max\{|a_i| - \max_i |a_i| + \epsilon/d, 0\})| \leq \sum_{i=1}^{d}|P_{[\check{c}_i, \hat{c}_i]}(\text{sign}(a_i)(\epsilon/d))| \leq \sum_{i=1}^{d}(\epsilon/d) = \epsilon$.

$\ell_2$ **norm** When we find the Euclidean projection of $\mathbf{a}$ onto the set $\mathcal{X}$, we solve

$$\underset{\boldsymbol{\delta}}{\text{minimize}} \quad \|\boldsymbol{\delta} - \mathbf{a}\|_2^2 + I_{[\check{\mathbf{c}}, \hat{\mathbf{c}}]}(\boldsymbol{\delta}) \\ \text{subject to} \quad \|\boldsymbol{\delta}\|_2^2 \leq \epsilon^2, \tag{29}$$

where $I_{[\check{\mathbf{c}}, \hat{\mathbf{c}}]}(\cdot)$ is the indicator function of the set $[\check{\mathbf{c}}, \hat{\mathbf{c}}]$. The Langragian of this problem is

$$L = \|\boldsymbol{\delta} - \mathbf{a}\|_2^2 + I_{[\check{\mathbf{c}}, \hat{\mathbf{c}}]}(\boldsymbol{\delta}) + \lambda_2(\|\boldsymbol{\delta}\|_2^2 - \epsilon^2) \tag{30}$$

$$= \sum_{i=1}^{d}((\delta_i - a_i)^2 + \lambda_2\delta_i^2 + I_{[\check{c}_i, \hat{c}_i]}(\delta_i)) - \lambda_2\epsilon^2. \tag{31}$$

The minimizer $\boldsymbol{\delta}^*$ minimizes the Lagrangian, it is

$$\delta_i^* = P_{[\check{c}_i, \hat{c}_i]}(\frac{1}{\lambda_2 + 1}a_i).$$

The primal, dual feasibility and complementary slackness are

$$\lambda_2 = 0, \|\boldsymbol{\delta}\|_2^2 = \sum_{i=1}^{d}\delta_i^2 = \sum_{i=1}^{d}(P_{[\check{c}_i, \hat{c}_i]}(a_i))^2 \leq \epsilon^2 \tag{32}$$

$$\text{or } \lambda_2 > 0, \|\boldsymbol{\delta}\|_2^2 = \sum_{i=1}^{d}\delta_i^2 = (P_{[\check{c}_i, \hat{c}_i]}(\frac{1}{\lambda_2 + 1}a_i))^2 = \epsilon^2. \tag{33}$$

If $\sum_{i=1}^{d}(P_{[\check{c}_i,\hat{c}_i]}(a_i))^2 \le \epsilon^2$, $\delta_i^* = P_{[\check{c}_i,\hat{c}_i]}(a_i)$. Otherwise $\delta_i^* = P_{[\check{c}_i,\hat{c}_i]}\left(\frac{1}{\lambda_2+1}a_i\right)$, where $\lambda_2$ is given by the root of the equation $\sum_{i=1}^{d}(P_{[\check{c}_i,\hat{c}_i]}(\frac{1}{\lambda_2+1}a_i))^2 = \epsilon^2$. Bisection method can be used to solve the above equation for $\lambda_2$, starting with the initial interval $(0, \sqrt{\sum_{i=1}^{d}(a_i)^2}/\epsilon - 1]$. Since $\sum_{i=1}^{d}(P_{[\check{c}_i,\hat{c}_i]}(\frac{1}{0+1}a_i))^2 = \sum_{i=1}^{d}(P_{[\check{c}_i,\hat{c}_i]}(a_i))^2 > \epsilon^2$ in this case, and $\sum_{i=1}^{d}(P_{[\check{c}_i,\hat{c}_i]}(\frac{1}{\lambda_2+1}a_i))^2 = \sum_{i=1}^{d}(P_{[\check{c}_i,\hat{c}_i]}(\epsilon a_i/\sqrt{\sum_{i=1}^{d}(a_i)^2}))^2 \le \epsilon^2 \sum_{i=1}^{d}(a_i)^2/(\sqrt{\sum_{i=1}^{d}(a_i)^2})^2 = \epsilon^2$.

$\ell_0$ **norm** For $\ell_0$ norm in $\mathcal{X}$, it is independent to the box constraint. So we can clip $\mathbf{a}$ to the box constraint first, which is $\delta_i' = P_{[\check{c}_i,\hat{c}_i]}(a_i)$, and then project it onto $\ell_0$ norm.

We find the additional Euclidean distance of every element in $\mathbf{a}$ and zero after they are clipped to the box constraint, which is

$$\eta_i = \begin{cases} \sqrt{a_i^2 - (a_i - \check{c}_i)^2} & a_i < \check{c}_i \\ \sqrt{a_i^2 - (a_i - \hat{c}_i)^2} & a_i > \hat{c}_i \\ |a_i| & \text{otherwise.} \end{cases} \tag{34}$$

It can be equivalently written as

$$\eta_i = \begin{cases} \sqrt{2a_i\check{c}_i - \check{c}_i^2} & a_i < \check{c}_i \\ \sqrt{2a_i\hat{c}_i - \hat{c}_i^2} & a_i > \hat{c}_i \\ |a_i| & \text{otherwise.} \end{cases} \tag{35}$$

To derive the Euclidean projection onto $\ell_0$ norm, we find the $\epsilon$-th largest element in $\boldsymbol{\eta}$ and call it $[\boldsymbol{\eta}]_\epsilon$. We keep the elements whose corresponding $\eta_i$ is above or equals to $\epsilon$-th, and set rest to zeros. The closed-form solution is given by

$$\delta_i^* = \begin{cases} \delta_i' & \eta_i \ge [\boldsymbol{\eta}]_\epsilon \\ 0 & \text{otherwise.} \end{cases} \tag{36}$$

$\square$

*Difference to (Hein & Andriushchenko, 2017, Proposition 4.1).* We remark that Hein & Andriushchenko (2017) discussed a relevant problem of generating $\ell_p$-norm based adversarial examples under box and linearized classification constraints. It was shown in (Hein & Andriushchenko, 2017, Proposition 4.1) that the problem is convex and the solution can be derived using KKT conditions. However, Proposition 1 in our paper is different from (Hein & Andriushchenko, 2017, Proposition 4.1). First, we place $\ell_p$ norm as a hard constraint rather than minimizing it in the objective function. This difference will make our Lagrangian function more involved with a newly introduced non-negative Lagrangian multiplier. Second, the problem of our interest is projection onto the intersection of box and $\ell_p$ constraints. Such a projection step can then be combined with an attack loss (no need of linearization) for generating adversarial examples. Third, we cover the case of $\ell_0$ norm.

## C EXPERIMENT SETUP

### C.1 MODEL ARCHITECTURES AND TRAINING DETAILS

For a comprehensive evaluation of proposed algorithms, we adopt a set of diverse DNN models (Model A to H), including multi-layer perceptrons (MLP), All-CNNs Springenberg et al. (2015), LeNet Lecun et al. (1998), LeNetV2[3], VGG16 Simonyan & Zisserman (2015), ResNet50 He et al. (2016), Wide-ResNet Madry et al. (2017) and GoogLeNet Szegedy et al. (2015). For the last four models, we use the exact same architecture as original papers and evaluate them only on CIFAR-10 dataset. The details for model architectures are provided in Table A1. For compatibility with our framework, we implement and train these models based on the strategies adopted in pytorch-cifar[4] and achieve comparable performance on clean images; see Table A2. To foster reproducibility, all the trained models are publicly accessible in the anonymous link. Specifically, we trained MNIST

---

[3] An enhanced version of original LeNet with more layers and units (see Table A1 Model D).

[4] `https://github.com/kuangliu/pytorch-cifar`

classifiers for 50 epochs with Adam and a constant learning rate of 0.001. For CIFAR-10 classifers, the models are trained for 250 epochs with SGD (using 0.8 nesterov momentum, weight decay $5e^{-4}$). The learning rate is reduced at epoch 100 and 175 with a decay rate of 0.1. The initial learning rate is set as 0.01 for models {A, B, C, D, H} and 0.1 for {E, F, G}. Note that no data augmentation is employed in the training.

**Table A1:** Neural network architectures used on the MNIST and CIFAR-10 dataset. Conv: convolutional layer, FC: fully connected layer, Globalpool: global average pooling layer.

| **A** (MLP) | **B** (All-CNNs, 2015) | **C** (LeNet, 1998) | **D** (LeNetV2) |
|---|---|---|---|
| FC(128) + Relu | Conv([32, 64], 3, 3) + Relu | Conv(6, 5, 5) + Relu | Conv(32, 3, 3) + Relu |
| FC(128) + Relu | Conv(128, 3, 3) + Dropout(0.5) | Maxpool(2, 2) | Maxpool(2, 2) |
| FC(64) + Relu | Conv([128, 128], 3, 3) + Relu | Conv(16, 5, 5) + Relu | Conv(64, 3, 3) + Relu |
| FC(10) | Conv(128, 3, 3) + Dropout(0.5) | Maxpool(2, 2) | Maxpool(2, 2) |
| Softmax | Conv(128, 3, 3) + Relu | FC(120) + Relu | FC(128) + Relu |
| | Conv(128, 1, 1) + Relu | FC(84) + Relu | Dropout(0.25) |
| | Conv(10, 1, 1) + Globalpool | FC(10) | FC(10) |
| | Softmax | Softmax | Softmax |
| **E** (VGG16, 2015) | **F** (ResNet50, 2016) | **G** (Wide-ResNet, 2017) | **H** (GoogLeNet, 2015) |

**Table A2:** Clean test accuracy of DNN models on MNIST and CIFAR-10. We roughly derive the model robustness by attacking models separately using FGSM Goodfellow et al. (2014). The adversarial examples are generated by FGSM $\ell_\infty$-attack ($\epsilon = 0.2$).

| MNIST | | | CIFAR-10 | | | | | |
|---|---|---|---|---|---|---|---|---|
| Model | Acc. | FGSM | Model | Acc. | FGSM | Model | Acc. | FGSM |
| A: MLP | 98.20% | 18.92% | A: MLP | 55.36% | 11.25% | E: VGG16 | 87.57% | 10.83% |
| B: All-CNNs | 99.49% | 50.95% | B: All-CNNs | 84.18% | 9.89% | F: ResNet50 | 88.11% | 10.73% |
| C: LeNet | 99.25% | 63.23% | C: LeNet | 64.95% | 14.45% | G: Wide-ResNet | 91.67% | 15.78% |
| D: LeNetV2 | 99.33% | 56.36% | D: LeNetV2 | 74.89% | 9.77% | H: GoogLeNet | 90.92% | 9.91% |

## C.2 CRAFTING ADVERSARIAL EXAMPLES

We adopt variant C&W loss in APGD/PGD as suggested in Madry et al. (2017); Carlini & Wagner (2017) with a confidence parameter $\kappa = 50$. Cross-entropy loss is also supported in our implementation. The adversarial examples are generated by 20-step PGD/APGD unless otherwise stated (e.g., 50 steps for ensemble attacks). Note that proposed algorithms are robust and will not be affected largely by the choices of hyperparameters ($\alpha, \beta, \gamma$). In consequence, we do not finely tune the parameters on the validation set. Specifically, The learning rates $\alpha, \beta$ and regularization factor $\gamma$ for Table 1 are set as - (a) MNIST: $\ell_0 : \alpha = 1, \beta = \frac{1}{100}, \gamma = 7$, $\ell_1 : \alpha = \frac{1}{4}, \beta = \frac{1}{100}, \gamma = 5$, $\ell_2 : \alpha = \frac{1}{10}, \beta = \frac{1}{100}, \gamma = 3$; $\ell_\infty : \alpha = \frac{1}{4}, \beta = \frac{1}{50}, \gamma = 3$; (b) CIFAR-10: $\ell_0 : \alpha = 1, \beta = \frac{1}{150}, \gamma = 1$, $\ell_1 : \alpha = \frac{1}{4}, \beta = \frac{1}{100}, \gamma = 5$, $\ell_2 : \alpha = \frac{1}{8}, \beta = \frac{1}{100}, \gamma = 3$; $\ell_\infty : \alpha = \frac{1}{5}, \beta = \frac{1}{50}, \gamma = 6$.

Due to varying model robustness on different datasets, the perturbation magnitudes $\epsilon$ are set separately Carlini et al. (2019). For universal perturbation experiments, the $\epsilon$ are set as 0.2 (A, B), 0.3 (C) and 0.25 (D) on MNIST; 0.02 (B, H), 0.35 (E) and 0.05 (D) on CIFAR-10. For generalized AT, the models on MNIST are trained following the same rules in last section, except that training epochs are prolonged to 350 and adversarial examples are crafted for assisting the training with a ratio of 0.5. Our experiment setup is based on CleverHans package[5] and Carlini and Wagner's framework[6].

## C.3 DETAILS OF CONDUCTED DATA TRANSFORMATIONS

To demonstrate the effectiveness of APGD in generating robust adversarial examples against multiple transformations, we adopt a series of common transformations, including a&b) flipping images

---

[5] https://github.com/tensorflow/cleverhans
[6] https://github.com/carlini/nn_robust_attacks

horizontally (*flh*) and vertically (*flv*); c) adjusting image brightness (*bri*); d) performing gamma correction (*gam*), e) cropping and re-sizing images (*crop*); f) rotating images (*rot*).

Moreover, both deterministic and stochastic transformations are considered in our experiments. In particular, Table 3 and Table A6 are deterministic settings - *rot*: rotating images 30 degree clockwise; *crop*: cropping images in the center ($0.8 \times 0.8$) and resizing them to $32 \times 32$; *bri*: adjusting the brightness of images with a scale of 0.1; *gam*: performing gamma correction with a value of 1.3. Differently, in Table A5, we introduce randomness for drawing samples from the distribution - *rot*: rotating images randomly from -10 to 10 degree; *crop*: cropping images in the center randomly (from 0.6 to 1.0); other transformations are done with a probability of 0.8. In experiments, we adopt `tf.image` API [7] for processing the images.

# D    SUPPLEMENTARY RESULTS - ROBUST ADVERSARIAL ATTACKS

## D.1    ENSEMBLE ATTACK OVER MULTIPLE MODELS

Table A3 shows the performance of average (ensemble PGD Liu et al. (2018)) and min-max (APGD) strategies for attacking model ensembles. Our min-max approach results in 15.69% averaged improvement on $\text{ASR}_{all}$ over models {A, E, F, H} on CIFAR-10.

**Table A3:** Comparison of average and min-max (APGD) ensemble attack over four models on CIFAR-10. Acc (%) represents the test accuracy of classifiers on adversarial examples. The learning rates $\alpha, \beta$ and regularization factor $\gamma$ are set as - $\ell_0 : \alpha = 1, \beta = \frac{1}{150}, \gamma = 1, \ell_1 : \alpha = \frac{1}{4}, \beta = \frac{1}{100}, \gamma = 5, \ell_2 : \alpha = \frac{1}{8}, \beta = \frac{1}{100}, \gamma = 3;$ $\ell_\infty : \alpha = \frac{1}{5}, \beta = \frac{1}{50}, \gamma = 6$. The attack iteration for APGD is set as 50.

| Box constraint | Opt. | $\text{Acc}_A$ | $\text{Acc}_E$ | $\text{Acc}_F$ | $\text{Acc}_H$ | $\text{ASR}_{all}$ | Lift ($\uparrow$) |
|---|---|---|---|---|---|---|---|
| $\ell_0$ ($\epsilon = 70$) | *avg.* | 27.38 | 6.33 | 7.18 | 6.99 | 66.56 | - |
| | min max | 19.38 | 8.72 | 9.48 | 8.94 | **73.83** | **10.92%** |
| $\ell_1$ ($\epsilon = 30$) | *avg.* | 30.90 | 2.06 | 1.85 | 1.84 | 66.23 | - |
| | min max | 12.56 | 3.21 | 2.70 | 2.72 | **83.13** | **25.52%** |
| $\ell_2$ ($\epsilon = 1.5$) | *avg.* | 20.87 | 1.75 | 1.21 | 1.54 | 76.41 | - |
| | min max | 10.26 | 3.15 | 2.24 | 2.37 | **84.99** | **11.23%** |
| $\ell_\infty$ ($\epsilon = 0.03$) | *avg.* | 25.75 | 2.59 | 1.66 | 2.27 | 70.54 | - |
| | min max | 13.47 | 3.79 | 3.15 | 3.48 | **81.17** | **15.07%** |

## D.2    COMPARISON WITH HEURISTIC WEIGHTING SCHEMES

To further demonstrate the effectiveness of self-adjusted weighting factors in proposed min-max framework, we compare with heuristic weighting schemes in Table A4. Specifically, with the prior knowledge of robustness of given models (C > D > A > B), we devised several heuristic baselines including: (a) $w_{c+d}$: ensemble PGD on models C and D only; (b) $w_{a+c+d}$: ensemble PGD on models A, C and D only; (c) $w_{clip}$: clipped version of C&W loss (threshold $\beta = 40$) to balance model weights in optimization as suggested in Shafahi et al. (2018); (d) $w_{prior}$: larger weights on the more robust models, $w_{prior} = [w_A, w_B, w_C, w_D] = [0.2, 0.1, 0.4, 0.3]$; (e) $w_{static}$: the converged mean weights of min-max (APGD) ensemble attack. For $\ell_2$ ($\epsilon = 3.0$) and $\ell_\infty$ ($\epsilon = 0.2$) attacks, $w_{static} = [w_A, w_B, w_C, w_D]$ are $[0.209, 0.046, 0.495, 0.250]$ and $[0.080, 0.076, 0.541, 0.303]$, respectively.

Table A4 shows that our min-max approach outperforms all static heuristic weighting schemes by a large margin. Specifically, our min-max APGD also achieves significant improvement compared to $w_{static}$ setting, where the converged optimal weights are statically (i.e., invariant w.r.t different images and attack procedure) adopted. It again verifies the benefits of proposed min-max approach by automatically learning the weights for different examples during the process of ensemble attack generation (see Figure 1c).

---

[7] https://www.tensorflow.org/api_docs/python/tf/image

**Table A4:** Comparison of average, min-max (APGD) ensemble attack and some heuristic weighting schemes over four models on MNIST. Acc (%) represents the test accuracy of classifiers on adversarial examples. $w_{c+d}$ and $w_{a+c+d}$: average ensemble attack over models {C, D} and {A, C, D}; $w_{clip}$: clipped version of C&W loss (Shafahi et al., 2018). $w_{prior}$: larger weights on the models that are more difficult to attack - $[w_A, w_B, w_C, w_D] = [0.2, 0.1, 0.4, 0.3]$; $w_{static}$: converged mean weights (over all data) in $\min\max$ ensemble attack. The experimental setting is the same as Table 1.

| Box constraint | Opt. | $Acc_A$ | $Acc_B$ | $Acc_C$ | $Acc_D$ | $ASR_{avg}$ | $ASR_{all}$ | Lift ($\uparrow$) |
|---|---|---|---|---|---|---|---|---|
| | $avg.$ | 6.88 | 0.03 | 26.28 | 14.50 | 88.08 | 69.12 | - |
| | $w_{c+d}$ | 69.03 | 14.58 | 5.11 | 0.34 | 77.74 | 28.65 | -58.56% |
| | $w_{a+c+d}$ | 1.34 | 24.53 | 11.69 | 2.79 | 89.91 | 67.45 | -2.42% |
| $\ell_2$ ($\epsilon = 3.0$) | $w_{clip}$ | 2.70 | 0.02 | 12.69 | 4.13 | 95.12 | 85.33 | 23.45% |
| | $w_{prior}$ | 6.28 | 0.05 | 6.78 | 2.65 | 96.06 | 88.25 | 27.68% |
| | $w_{static}$ | 4.52 | 0.27 | 3.35 | 4.15 | 96.93 | 90.53 | 30.98% |
| | $\min\max$ | 1.51 | 0.89 | 3.50 | 2.06 | **98.01** | **95.31** | **37.89%** |
| | $avg.$ | 1.05 | 0.07 | 41.10 | 35.03 | 80.69 | 48.17 | - |
| | $w_{c+d}$ | 60.37 | 19.55 | 15.10 | 1.87 | 75.78 | 29.32 | -39.13% |
| | $w_{a+c+d}$ | 0.46 | 21.57 | 25.36 | 13.84 | 84.69 | 53.39 | 10.84% |
| $\ell_\infty$ ($\epsilon = 0.2$) | $w_{clip}$ | 0.66 | 0.03 | 23.43 | 13.23 | 90.66 | 71.54 | 48.52% |
| | $w_{prior}$ | 1.57 | 0.24 | 17.67 | 13.74 | 91.70 | 74.34 | 54.33% |
| | $w_{static}$ | 10.58 | 0.39 | 9.28 | 10.05 | 92.43 | 77.84 | 61.59% |
| | $\min\max$ | 2.47 | 0.37 | 7.39 | 5.81 | **95.99** | **90.16** | **87.17%** |

## D.3 ROBUST ADVERSARIAL ATTACK OVER DATA TRANSFORMATIONS

Table A5 and A6 compare the performance of average (EOT Athalye et al. (2018a)) and min-max (APGD) strategies. Our approach results in 4.31% and 8.22% averaged lift over four models {A, B, C, D} on CIFAR-10 under given stochastic and deterministic transformation sets.

**Table A5:** Comparison of average and min-max optimization on robust attack over multiple data transformations on CIFAR-10. Note that all data transformations are conducted stochastically with a probability of 0.8, except for *crop* which randomly crops a central area from original image and re-size it into $32 \times 32$. The adversarial examples are generated by 20-step $\ell_\infty$-APGD ($\epsilon = 0.03$) with $\alpha = \frac{1}{2}, \beta = \frac{1}{100}$ and $\gamma = 10$.

| Model | Opt. | $Acc_{ori}$ | $Acc_{flh}$ | $Acc_{flv}$ | $Acc_{bri}$ | $Acc_{crop}$ | $ASR_{avg}$ | $ASR_{gp}$ | Lift ($\uparrow$) |
|---|---|---|---|---|---|---|---|---|---|
| A | $avg.$ | 11.55 | 21.60 | 13.64 | 12.30 | 22.37 | 83.71 | 55.97 | - |
| | $\min\max$ | 13.06 | 18.90 | 13.43 | 13.90 | 20.27 | 84.09 | **59.17** | **5.72%** |
| B | $avg.$ | 6.74 | 11.55 | 10.33 | 6.59 | 18.21 | 89.32 | 69.52 | - |
| | $\min\max$ | 8.19 | 11.13 | 10.31 | 8.31 | 16.29 | 89.15 | **71.18** | **2.39%** |
| C | $avg.$ | 8.23 | 17.47 | 13.93 | 8.54 | 18.83 | 86.60 | 58.85 | - |
| | $\min\max$ | 9.68 | 13.45 | 13.41 | 9.95 | 18.23 | 87.06 | **61.63** | **4.72%** |
| D | $avg.$ | 8.67 | 19.75 | 11.60 | 8.46 | 19.35 | 86.43 | 60.96 | - |
| | $\min\max$ | 10.43 | 16.41 | 12.14 | 10.15 | 17.64 | 86.65 | **63.64** | **4.40%** |

**Table A6:** Comparison of average and min-max optimization on robust attack over multiple data transformations on CIFAR-10. Here a new rotation (*rot*) transformation is introduced, where images are rotated 30 degrees clockwise. Note that all data transformations are conducted with a probability of 1.0. The adversarial examples are generated by 20-step $\ell_\infty$-APGD ($\epsilon = 0.03$) with $\alpha = \frac{1}{2}, \beta = \frac{1}{100}$ and $\gamma = 10$.

| Model | Opt. | $Acc_{ori}$ | $Acc_{flh}$ | $Acc_{flv}$ | $Acc_{bri}$ | $Acc_{gam}$ | $Acc_{crop}$ | $Acc_{rot}$ | $ASR_{avg}$ | $ASR_{gp}$ | Lift ($\uparrow$) |
|---|---|---|---|---|---|---|---|---|---|---|---|
| A | $avg.$ | 11.06 | 22.37 | 14.81 | 12.32 | 10.92 | 20.40 | 15.89 | 84.60 | 49.24 | - |
| | $\min\max$ | 13.51 | 18.84 | 14.03 | 15.20 | 13.00 | 18.03 | 14.79 | 84.66 | **52.31** | **6.23%** |
| B | $avg.$ | 5.55 | 11.96 | 9.97 | 5.63 | 5.94 | 16.42 | 11.47 | 90.44 | 65.18 | - |
| | $\min\max$ | 6.75 | 9.13 | 10.56 | 6.72 | 7.11 | 12.23 | 10.80 | 90.96 | **70.38** | **7.98%** |
| C | $avg.$ | 7.65 | 22.30 | 15.82 | 8.17 | 8.07 | 15.44 | 15.09 | 86.78 | 49.67 | - |
| | $\min\max$ | 9.05 | 15.10 | 14.57 | 9.57 | 9.31 | 14.11 | 14.23 | 87.72 | **55.37** | **11.48%** |
| D | $avg.$ | 8.22 | 20.88 | 13.49 | 7.91 | 8.71 | 16.33 | 14.98 | 87.07 | 53.52 | - |
| | $\min\max$ | 10.17 | 14.65 | 13.62 | 10.03 | 10.35 | 14.36 | 13.82 | 87.57 | **57.36** | **7.17%** |

## D.4 SENSITIVITY ANALYSIS OF REGULARIZER ON PROBABILITY SIMPLEX

To further explore the utility of quadratic regularizer on the probability simplex in proposed min-max framework, we conducted sensitivity analysis on $\gamma$ and show how the proposed regularization affects the eventual performance (Figure A1) taking ensemble attack as an example. The experimental setting is the same as Table 1 except for altering the value of $\gamma$ from 0 to 10. Figure A1 shows that too small or too large $\gamma$ leads to relative weak performance due to the unstable convergence and penalizing too much for average case. When $\gamma$ is around 4, APGD will achieve the best performance so we adopted this value in the experiments (Table 1). Moreover, when $\gamma \to \infty$, the regularizer term dominates the optimization objective and it becomes the average case.

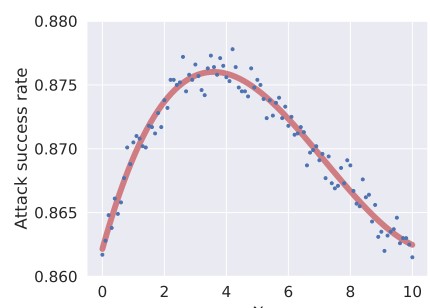

**Figure A1:** Sensitivity analysis of the regularizer $\frac{\gamma}{2}\|\mathbf{w} - \mathbf{1}/K\|_2^2$ on the probability simplex.

## E    SUPPLEMENTARY RESULTS - ADVERSARIAL TRAINING AGAINST MULTIPLE TYPES OF ADVERSARIAL ATTACKS

### E.1    ROBUSTNESS EVALUATION

Figure A2 presents "overall robustness" comparison of our min-max generalized AT scheme and vanilla AT with single type of attacks ($\ell_\infty$ and $\ell_2$) on MNIST (LeNet). Similarly, our min-max training scheme leads to a higher "overall robustness" measured by $S_\epsilon$. In practice, due to the lacking knowledge of the strengths/types of the attacks used by adversaries, it is meaningful to enhance "overall robustness" of models under the worst perturbation ($\text{Acc}_{\text{adv}}^{\text{max}}$). Specifically, our min-max generalized AT leads to 6.27% and 17.63% improvement on $S_\epsilon$ compared to single-type AT with $\ell_\infty$ and $\ell_2$ attacks. Furthermore, weighting factor $w$ of the probability simplex helps understand the behavior of AT under mixed types of attacks. Our AMPGD algorithm will adjust $w$ automatically according to the min-max principle - defending the strongest attack. In Figure A2a, as $\epsilon_{\ell_2}$ increases, $\ell_2$-attack becomes stronger so its corresponding $w$ increases as well. When $\epsilon_{\ell_2} \geq 2.5$, $\ell_2$-attack dominates the adversarial training process. That is to say, our AMPGD algorithm will put more weights on stronger attacks even if the strengths of attacks are unknown, which is a meritorious feature in practice.

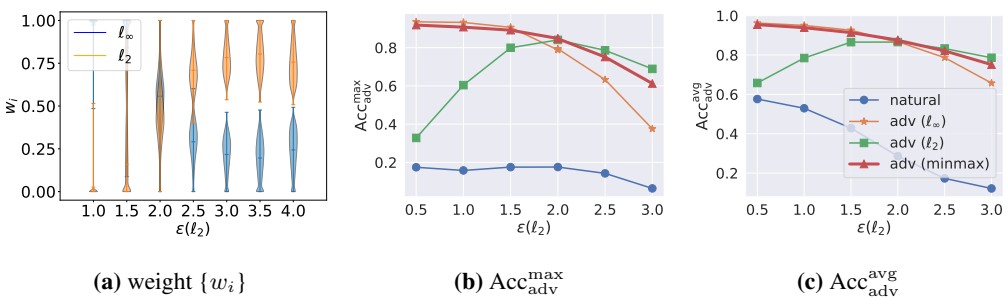

| **(a)** weight $\{w_i\}$ | **(b)** $\text{Acc}_{\text{adv}}^{\text{max}}$ | **(c)** $\text{Acc}_{\text{adv}}^{\text{avg}}$ |
| --- | --- | --- |

**Figure A2:** (a): Violin plot of weight $w$ in APGD as a function of perturbation magnitude $\epsilon$ of $\ell_2$ attack in adversarial training; (b) & (c): Robustness of LeNet (Model C) under different adversarial training schemes.

Table A8 shows complete results on the test accuracy of models in different training schemes. In general, the min-max generalized AT obtains better performance than averaging strategy. AMPGD always leads to Top-2 $\text{Acc}_{\text{adv}}^{\text{max}}$ and $\text{Acc}_{\text{adv}}^{\text{avg}}$.

### E.2    COMPARISON WITH UNIVERSAL ADVERSARIAL TRAINING (UAT)

Shafahi et al. (2018) also propose a variant of adversarial training to defend universal perturbations over multiple images. To produce universal perturbations, they propose uSGD to conduct gradient descent on the averaged loss of one-batch images. In consequence, their approach can be regarded as

a variant of our generalized AT in average case. The difference is that they do AT across multiple adversarial images under universal perturbation rather than mixed $\ell_p$-norm perturbations.

We added UAT [1] as one of our defense baselines in Table A7. The universal perturbation is generated by uSGD ($\ell_\infty$ norm, $\epsilon = 0.3$) with a batch size of 128 following Shafahi et al. (2018). We find that a) our proposed approach outperforms UAT under per-image $\ell_p$ attacks. Taking A7a as an example, our $avg$ and $\min\max$ generalized AT (with DPAR) result in average 17.85% and 17.97% improvement in adversarial test accuracy (ATA), b) our approach has just 3.72% degradation in ATA when encountering universal attacks, and c) both methods yield very similar normal test accuracy. It is not surprising that our average and min-max training schemes can achieve better overall robustness while maintaining competitive performance on defending universal perturbation. This is because the defensed model is trained under more general ($\ell_p$ norm) and diversity promoted perturbations. As a result, proposed generalized AT is expected to obtain better overall robustness and higher transferability as shown in Table 4 and A7.

**Table A7:** Adversarial training of MNIST models on single attacks ($\ell_\infty$ and $\ell_2$), multiple attacks ($avg.$ and $\min\max$) and universal perturbation ($uni$). The perturbation magnitude $\epsilon$ for $\ell_\infty$ and $\ell_2$ attacks are 0.2 and 2.0, respectively. uSGD indicates universal adversarial training following Shafahi et al. (2018) ($\ell_\infty$ norm, $\epsilon = 0.3$). Top 2 test accuracy on each metric are highlighted.

**(a) MLP**

| Opt. | Acc. | Acc-$\ell_\infty$ | Acc-$\ell_2$ | Acc-$uni$ | Acc$_{adv}^{avg}$ |
|---|---|---|---|---|---|
| natural | 98.30 | 2.70 | 13.86 | 21.61 | 12.72 |
| $\ell_\infty$ | 98.08 | **77.70** | 69.17 | **90.90** | 79.26 |
| $\ell_2$ | 98.72 | 70.03 | **81.74** | 82.49 | 78.09 |
| uSGD ($\ell_\infty$) | 98.73 | 56.21 | 64.51 | **91.27** | 70.66 |
| $avg.$ | 98.62 | 75.09 | 79.00 | 86.69 | 80.26 |
| + DPAR | 98.50 | 76.75 | 79.67 | 87.88 | **81.44** |
| $\min\max$ | 98.59 | 75.96 | 79.15 | 86.13 | 80.41 |
| + DPAR | 98.58 | **76.92** | **79.74** | 87.55 | **81.40** |

**(b) LeNet**

| Opt. | Acc. | Acc-$\ell_\infty$ | Acc-$\ell_2$ | Acc-$uni$ | Acc$_{adv}^{avg}$ |
|---|---|---|---|---|---|
| natural | 99.25 | 17.93 | 39.32 | 58.93 | 38.73 |
| $\ell_\infty$ | 99.18 | **93.80** | 78.97 | **98.70** | 90.49 |
| $\ell_2$ | 99.22 | 85.84 | **87.31** | 96.63 | 89.93 |
| uSGD ($\ell_\infty$) | 99.44 | 72.81 | 66.39 | **98.37** | 79.19 |
| $avg.$ | 99.22 | 88.96 | 85.59 | 97.41 | 90.65 |
| + DPAR | 99.25 | 89.96 | **86.49** | 97.36 | **91.27** |
| $\min\max$ | 99.32 | 89.21 | 85.98 | 98.22 | 91.14 |
| + DPAR | 99.22 | **90.19** | 86.47 | 97.77 | **91.48** |

### E.3 OVERLAP OF $\ell_p$-NORM BALLS

As reported in Sec. 4.2, our min-max generalized AT does not always result in the best performance on the success rate of defending the worst/strongest perturbation (Acc$_{adv}^{max}$) for given $\left(\epsilon_{\ell_\infty}, \epsilon_{\ell_2}\right)$ pair, especially when the strengths of two attacks diverge greatly (e.g., $\epsilon$ for $\ell_\infty$ and $\ell_2$ attacks are 0.2 and 0.5). In what follows, we provide explanation and analysis about this finding inspired by recent work Araujo et al. (2019).

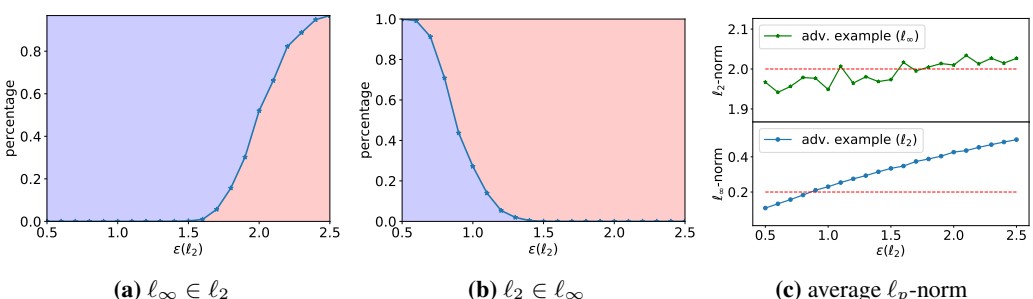

**(a)** $\ell_\infty \in \ell_2$          **(b)** $\ell_2 \in \ell_\infty$          **(c)** average $\ell_p$-norm

**Figure A3:** (a) & (b): Comparison of the percentage of adversarial examples inside $\ell_\infty$ ball (left, blue area) and inside $\ell_2$ ball (right, red area). In particular, the red (blue) area in (a) (or (b)) represents the percentage of adversarial examples crafted by $\ell_\infty$ ($\ell_2$) attack that also belong to $\ell_2$ ($\ell_\infty$) ball. We generate adversarial examples on 10,000 test images for each attack. (c): Average $\ell_p$ norm of adversarial examples as a function of perturbation magnitude $\epsilon_{\ell_2}$. The top (bottom) side represents the $\ell_2$-norm ($\ell_\infty$) of the adversarial examples generated by $\ell_\infty$ ($\ell_2$) attack as $\epsilon_{\ell_2}$ for generalized AT increases. Note that the same $\epsilon$ as the AT procedure is used while attacking trained robust models.

Figure A3 shows the real overlap of $\ell_\infty$ and $\ell_2$ norm balls in adversarial attacks for MLP model on MNIST. Ideally, if $\epsilon_{\ell_2}$ satisfies $\epsilon_{\ell_\infty} < \epsilon_{\ell_2} < \epsilon_{\ell_\infty} \times \sqrt{d}$, $\ell_\infty$ and $\ell_2$ balls will not cover each other

completely Araujo et al. (2019). In other words, AT with $\ell_\infty$ and $\ell_2$ attacks cannot interchange with each other. However, the real range of $\epsilon_{\ell_2}$ for keeping $\ell_2$ and $\ell_\infty$ balls intersected is not $(\epsilon_{\ell_\infty}, \epsilon_{\ell_\infty} \times \sqrt{d})$, because crafted adversarial examples are not uniformly distributed in $\ell_p$-norm balls. In Figure A3b, 99.98% adversarial examples devising using $\ell_2$ attack are also inside $\ell_\infty$ ball, even if $0.2 < \epsilon_{\ell_2} = 0.5 < 5.6$. In consequence, AT with $\ell_\infty$ attack is enough to handle $\ell_2$-attack in overwhelming majority cases, which results in better performance than min-max optimization (Table A8a).

Figure A3c presents the average $\ell_p$ distance of adversarial examples with $\epsilon_{\ell_2}$ increasing. The average $\ell_2$-norm (green line) of adversarial examples generated by $\ell_\infty$ attack remains around 2.0 with a slight rising trend. This is consistent to our setting - fixing $\epsilon_{\ell_2}$ as 0.2. It also indicates model robustness may effect the behavior of attacks - as $\epsilon_{\ell_2}$ increases, robustly trained MLP model becomes more robust against $\ell_2$ examples, so the $\ell_\infty$ attacker implicitly increases $\ell_2$ norm to attack the model more effectively. On the other hand, the average $\ell_\infty$-norm increases substantially as $\epsilon_{\ell_2}$ increases from 0.5 to 2.5. When $\epsilon_{\ell_2}$ arriving at 0.85, the average $\ell_\infty$ norm gets close to 0.2, so around half adversarial examples generated by $\ell_2$-attack are also inside $\ell_\infty$ balls, which is consistent with Table A3b.

### E.4 LEARNING CURVE UNDER DIFFERENT TRAINING SCHEMES

Figure A4 shows the learning curves of model A under different AT schemes, where two setting are plotted: (a) $(\epsilon_{\ell_\infty}, \epsilon_{\ell_2}) = (0.2, 0.5)$; (b) $(\epsilon_{\ell_\infty}, \epsilon_{\ell_2}) = (0.2, 2.0)$. Apart from better worst-case robustness shown in Table A8, our min-max generalized AT leads to a faster convergence compared to average-based AT, especially when the strengths of two attacks diverge greatly. For instance, when $\epsilon_{\ell_2} = 0.5$ (Figure A4a), the robust model trained with AMPGD reaches 70% test accuracy on the worst perturbation (1-$\mathcal{R}_{\text{adv}}^{\max}$) within 210 epochs versus 280 epochs in average setting. When $\epsilon_{\ell_2} = 2.0$ (Figure A4b), the learning curves for min-max and average strategy are very close because the strengths of two attacks are similar, which is verified by approximately equal weights in Figure 3a.

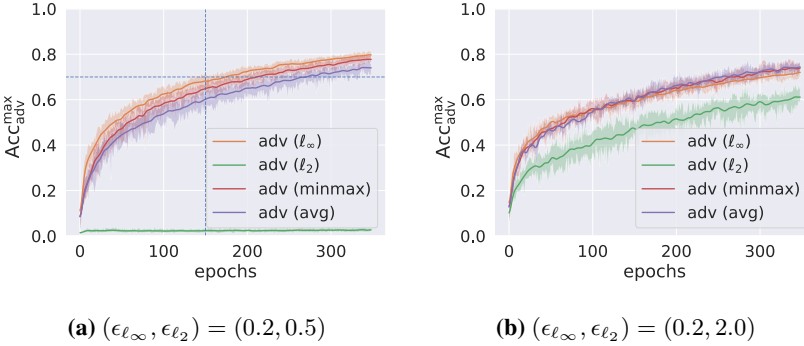

**(a)** $(\epsilon_{\ell_\infty}, \epsilon_{\ell_2}) = (0.2, 0.5)$          **(b)** $(\epsilon_{\ell_\infty}, \epsilon_{\ell_2}) = (0.2, 2.0)$

**Figure A4:** Learning curves of MLP model under different adversarial training schemes on MNIST. Note that each experiment is repeated ten times with different random seeds.

## F  INTERPRETABILITY OF DOMAIN WEIGHT $w$ ON UNIVERSAL PERTURBATION TO MULTIPLE IMAGES

Tracking *domain weight* $w$ of the probability simplex from our algorithms is an exclusive feature of solving problem 1. In Sec. 4, we show the strength of $w$ in understanding the procedure of optimization and interpreting the adversarial robustness. Here we would like to show the usage of $w$ in measuring "image robustness" on devising universal perturbation to multiple input samples. Table A9 and A10 show the image groups on MNIST with weight $w$ in APGD and two metrics (distortion of $\ell2$-C&W, minimum $\epsilon$ for $\ell_\infty$-PGD) of measuring the difficulty of attacking single images. The binary search is utilized to searching for the minimum perturbation.

Although adversaries need to consider a trade-off between multiple images while devising universal perturbation, we find that weighting factor $w$ in APGD is highly correlated under different $\ell_p$ norms. Furthermore, $w$ is also highly related to minimum distortion required for attacking a single image

**Table A8:** Adversarial training of MNIST models with single attacks ($\ell_\infty$ and $\ell_2$) and multiple attacks ($avg.$ and $\min\max$). During the training process, the perturbation magnitude $\epsilon_{\ell_\infty}$ is fixed as 0.2, and $\epsilon_{\ell_2}$ are changed from 0.5 to 3.0 with a step size of 0.5. For min-max scheme, the adversarial examples are crafted using 20-step $\ell_\infty$-APGD with $\alpha = \frac{1}{6}, \beta = \frac{1}{50}$ and $\gamma = 4$. The ratio of adversarial and benign examples in adversarial training is set as 1.0. For diversity-promoting attack regularizer (DPAR) in generalized AT, the hyperparameter $\lambda = 0.1$.

**(a)** $(\epsilon_{\ell_\infty}, \epsilon_{\ell_2}) = (0.2, 0.5)$

| Model | Opt. | Acc. | Acc-$\ell_\infty$ | Acc-$\ell_2$ | Acc$_{adv}^{max}$ | Acc$_{adv}^{avg}$ |
|---|---|---|---|---|---|---|
| MLP | natural | 98.28 | 2.78 | 93.75 | 1.80 | 48.27 |
| | $\ell_\infty$ | 98.22 | **77.82** | **97.11** | **77.23** | **87.46** |
| | $\ell_2$ | 98.71 | 12.04 | 97.10 | 11.73 | 54.57 |
| | $avg.$ | 98.83 | 74.07 | 97.70 | 73.67 | 85.88 |
| | + DPAR | 98.56 | **77.32** | 97.74 | **76.98** | **87.53** |
| | $\min\max$ | 98.73 | 75.88 | 97.43 | 75.56 | 86.66 |
| | + DPAR | 98.75 | 77.04 | **97.81** | 76.72 | 87.43 |
| LeNet | natural | 99.17 | 18.16 | 97.56 | 15.23 | 57.86 |
| | $\ell_\infty$ | 99.27 | **93.60** | 98.74 | **93.26** | **96.17** |
| | $\ell_2$ | 99.43 | 34.30 | 98.49 | 26.89 | 66.39 |
| | $avg.$ | 99.29 | 90.69 | **98.89** | 90.34 | 94.79 |
| | + DPAR | 99.28 | 91.81 | **98.87** | 91.52 | **95.34** |
| | $\min\max$ | 99.35 | 90.81 | 98.74 | 90.21 | 94.78 |
| | + DPAR | 99.34 | **91.82** | 98.77 | **91.60** | 95.30 |

**(b)** $(\epsilon_{\ell_\infty}, \epsilon_{\ell_2}) = (0.2, 1.0)$

| Model | Opt. | Acc. | Acc-$\ell_\infty$ | Acc-$\ell_2$ | Acc$_{adv}^{max}$ | Acc$_{adv}^{avg}$ |
|---|---|---|---|---|---|---|
| MLP | natural | 98.30 | 3.65 | 72.39 | 1.17 | 39.01 |
| | $\ell_\infty$ | 98.29 | **78.15** | 93.28 | **77.95** | **85.71** |
| | $\ell_2$ | 98.98 | 36.02 | 94.39 | 34.68 | 65.20 |
| | $avg.$ | 98.72 | 73.97 | **94.63** | 73.70 | 84.30 |
| | + DPAR | 98.60 | 76.57 | 94.41 | 76.39 | 85.49 |
| | $\min\max$ | 98.72 | 75.18 | 94.29 | 74.92 | 84.74 |
| | + DPAR | 98.68 | **76.59** | **95.11** | 76.49 | **85.85** |
| LeNet | natural | 9.16 | 18.24 | 89.97 | 15.36 | 54.10 |
| | $\ell_\infty$ | 99.28 | **93.51** | 96.49 | **93.13** | **95.00** |
| | $\ell_2$ | 99.50 | 63.48 | 96.62 | 57.94 | 80.05 |
| | $avg.$ | 99.40 | 89.39 | 96.94 | 89.02 | 93.16 |
| | + DPAR | 99.38 | 99.09 | **97.13** | 89.99 | 93.61 |
| | $\min\max$ | 99.31 | 90.82 | **97.20** | 90.56 | **94.01** |
| | + DPAR | 99.35 | **90.88** | 97.07 | **90.80** | 93.98 |

**(c)** $(\epsilon_{\ell_\infty}, \epsilon_{\ell_2}) = (0.2, 1.5)$

| Model | Opt. | Acc. | Acc-$\ell_\infty$ | Acc-$\ell_2$ | Acc$_{adv}^{max}$ | Acc$_{adv}^{avg}$ |
|---|---|---|---|---|---|---|
| MLP | natural | 98.39 | 2.77 | 35.70 | 2.32 | 19.23 |
| | $\ell_\infty$ | 98.34 | **78.96** | 85.94 | **77.42** | 82.45 |
| | $\ell_2$ | 99.00 | 60.37 | **89.96** | 59.82 | 75.16 |
| | $avg.$ | 98.61 | 75.01 | 88.85 | 74.76 | 81.93 |
| | + DPAR | 98.68 | 76.55 | 88.52 | 76.18 | **82.53** |
| | $\min\max$ | 98.76 | 75.66 | 88.78 | 75.33 | 82.22 |
| | + DPAR | 98.77 | **77.54** | **89.57** | **77.24** | **83.55** |
| LeNet | natural | 99.22 | 14.31 | 67.69 | 12.34 | 41.00 |
| | $\ell_\infty$ | 99.22 | **93.76** | 91.11 | **90.26** | **92.43** |
| | $\ell_2$ | 99.35 | 79.92 | 93.27 | 77.39 | 86.60 |
| | $avg.$ | 99.31 | 89.26 | **93.29** | 88.77 | 91.28 |
| | + DPAR | 99.27 | **90.75** | **93.48** | **89.96** | **92.11** |
| | $\min\max$ | 99.40 | 89.83 | 92.96 | 89.00 | 91.39 |
| | + DPAR | 99.35 | 90.64 | 93.27 | 89.80 | 91.96 |

**(d)** $(\epsilon_{\ell_\infty}, \epsilon_{\ell_2}) = (0.2, 2.0)$

| Model | Opt. | Acc. | Acc-$\ell_\infty$ | Acc-$\ell_2$ | Acc$_{adv}^{max}$ | Acc$_{adv}^{avg}$ |
|---|---|---|---|---|---|---|
| MLP | natural | 98.30 | 2.70 | 13.86 | 0.85 | 8.28 |
| | $\ell_\infty$ | 98.08 | **77.70** | 69.17 | 66.34 | 73.43 |
| | $\ell_2$ | 98.72 | 70.03 | **81.74** | 69.14 | 75.88 |
| | $avg.$ | 98.62 | 75.09 | 79.00 | 72.23 | 77.05 |
| | + DPAR | 98.50 | 76.75 | 79.67 | **74.14** | **78.21** |
| | $\min\max$ | 98.59 | 75.96 | 79.15 | 73.43 | 77.55 |
| | + DPAR | 98.58 | **76.92** | **79.74** | **74.29** | **78.35** |
| LeNet | natural | 99.25 | 17.93 | 39.32 | 17.57 | 28.63 |
| | $\ell_\infty$ | 99.18 | **93.80** | 78.97 | 78.80 | 86.39 |
| | $\ell_2$ | 99.22 | 85.84 | **87.31** | 84.06 | 86.58 |
| | $avg.$ | 99.22 | 88.96 | 85.59 | 84.29 | 87.28 |
| | + DPAR | 99.25 | 89.96 | 86.49 | 85.44 | **88.23** |
| | $\min\max$ | 99.32 | 89.21 | 85.98 | 84.82 | 87.60 |
| | + DPAR | 99.22 | **90.19** | 86.47 | **85.47** | 88.33 |

**(e)** $(\epsilon_{\ell_\infty}, \epsilon_{\ell_2}) = (0.2, 2.5)$

| Model | Opt. | Acc. | Acc-$\ell_\infty$ | Acc-$\ell_2$ | Acc$_{adv}^{max}$ | Acc$_{adv}^{avg}$ |
|---|---|---|---|---|---|---|
| MLP | natural | 98.31 | 3.37 | 6.02 | 2.27 | 4.70 |
| | $\ell_\infty$ | 98.25 | **77.91** | 51.28 | 49.40 | 64.59 |
| | $\ell_2$ | 98.10 | 73.94 | **70.01** | **67.66** | **71.97** |
| | $avg.$ | 98.47 | 75.35 | 64.39 | 63.37 | 69.86 |
| | + DPAR | 98.18 | 76.33 | **66.49** | 65.54 | 71.41 |
| | $\min\max$ | 98.44 | 75.48 | 66.12 | 64.99 | 70.80 |
| | + DPAR | 98.20 | **76.98** | 66.42 | **65.55** | **71.70** |
| LeNet | natural | 99.23 | 15.25 | 16.08 | 11.16 | 15.67 |
| | $\ell_\infty$ | 99.18 | **94.09** | 60.18 | 58.47 | 77.13 |
| | $\ell_2$ | 98.94 | 87.57 | **78.45** | **78.42** | **83.01** |
| | $avg.$ | 99.10 | 89.88 | 74.68 | 74.39 | 82.28 |
| | + DPAR | 99.14 | **90.17** | 75.16 | 75.09 | **82.67** |
| | $\min\max$ | 99.21 | 88.88 | 74.97 | 74.42 | 81.93 |
| | + DPAR | 99.09 | 89.34 | **75.55** | **75.45** | 82.45 |

**(f)** $(\epsilon_{\ell_\infty}, \epsilon_{\ell_2}) = (0.2, 3.0)$

| Model | Opt. | Acc. | Acc-$\ell_\infty$ | Acc-$\ell_2$ | Acc$_{adv}^{max}$ | Acc$_{adv}^{avg}$ |
|---|---|---|---|---|---|---|
| MLP | natural | 98.24 | 2.92 | 2.42 | 1.54 | 2.67 |
| | $\ell_\infty$ | 98.35 | **79.15** | 32.58 | 31.23 | 55.86 |
| | $\ell_2$ | 97.55 | 73.86 | **58.24** | **57.83** | **66.05** |
| | $avg.$ | 98.17 | 75.07 | 49.75 | 49.49 | 62.41 |
| | + DPAR | 97.85 | 74.61 | **51.16** | **51.04** | 62.89 |
| | $\min\max$ | 98.10 | 74.71 | 50.45 | 50.54 | 62.58 |
| | + DPAR | 97.97 | **76.13** | 51.12 | 51.00 | **63.63** |
| LeNet | natural | 99.24 | 13.76 | 4.74 | 2.57 | 9.25 |
| | $\ell_\infty$ | 99.30 | **93.14** | 39.48 | 32.93 | 65.81 |
| | $\ell_2$ | 98.55 | 87.87 | **68.69** | **68.34** | **78.28** |
| | $avg.$ | 99.10 | 89.19 | 59.87 | 60.01 | 74.53 |
| | + DPAR | 98.95 | **89.80** | 62.21 | 61.19 | 75.50 |
| | $\min\max$ | 99.01 | 88.93 | 61.15 | 60.76 | 75.04 |
| | + DPAR | 98.98 | 89.53 | **63.22** | **63.18** | 76.37 |

successfully. It means the inherent "image robustness" exists and effects the behavior of generating universal perturbation. Larger weight $w$ usually indicates an image with higher robustness (e.g., fifth 'zero' in the first row of Table A9), which usually corresponds to the MNIST letter with clear appearance (e.g., bold letter).

**Table A9:** Interpretability of domain weight $w$ for universal perturbation to multiple inputs on MNIST (*Digit 0 to 4*). Domain weight $w$ for different images under $\ell_p$-norm ($p = 0, 1, 2, \infty$) and two metrics measuring the difficulty of attacking single image are recorded, where dist. ($\ell_2$) denotes the the minimum distortion of successfully attacking images using C&W ($\ell_2$) attack; $\epsilon_{\min}$ ($\ell_\infty$) denotes the minimum perturbation magnitude for $\ell_\infty$-PGD attack.

| | Image | | | | | | | | | | |
|---|---|---|---|---|---|---|---|---|---|---|---|
| **Weight** | $\ell_0$ | 0. | 0. | 0. | 0. | 1.000 | 0.248 | 0.655 | 0.097 | 0. | 0. |
| | $\ell_1$ | 0. | 0. | 0. | 0. | 1.000 | 0.07 | 0.922 | 0. | 0. | 0. |
| | $\ell_2$ | 0. | 0. | 0. | 0. | 1.000 | 0.441 | 0.248 | 0.156 | 0.155 | 0. |
| | $\ell_\infty$ | 0. | 0. | 0. | 0. | 1.000 | 0.479 | 0.208 | 0.145 | 0.168 | 0. |
| **Metric** | dist.(C&W $\ell_2$) | 1.839 | 1.954 | 1.347 | 1.698 | 3.041 | 1.545 | 1.982 | 2.178 | 2.349 | 1.050 |
| | $\epsilon_{\min}$ ($\ell_\infty$) | 0.113 | 0.167 | 0.073 | 0.121 | 0.199 | 0.167 | 0.157 | 0.113 | 0.114 | 0.093 |
| | Image | | | | | | | | | | |
| **Weight** | $\ell_0$ | 0. | 0. | 0.613 | 0.180 | 0.206 | 0. | 0. | 0.223 | 0.440 | 0.337 |
| | $\ell_1$ | 0. | 0. | 0.298 | 0.376 | 0.327 | 0. | 0. | 0.397 | 0.433 | 0.169 |
| | $\ell_2$ | 0. | 0. | 0.387 | 0.367 | 0.246 | 0. | 0.242 | 0.310 | 0.195 | 0.253 |
| | $\ell_\infty$ | 0.087 | 0.142 | 0.277 | 0.247 | 0.246 | 0. | 0.342 | 0.001 | 0.144 | 0.514 |
| **Metric** | dist.(C&W $\ell_2$) | 1.090 | 1.182 | 1.327 | 1.458 | 0.943 | 0.113 | 1.113 | 1.357 | 1.474 | 1.197 |
| | $\epsilon_{\min}$ ($\ell_\infty$) | 0.075 | 0.068 | 0.091 | 0.105 | 0.096 | 0.015 | 0.090 | 0.076 | 0.095 | 0.106 |
| | Image | | | | | | | | | | |
| **Weight** | $\ell_0$ | 0. | 1.000 | 0. | 0. | 0. | 0. | 0. | 0.909 | 0. | 0.091 |
| | $\ell_1$ | 0. | 1.000 | 0. | 0. | 0. | 0. | 0. | 0.843 | 0. | 0.157 |
| | $\ell_2$ | 0. | 0.892 | 0. | 0. | 0.108 | 0. | 0. | 0.788 | 0. | 0.112 |
| | $\ell_\infty$ | 0. | 0.938 | 0. | 0. | 0.062 | 0. | 0. | 0.850 | 0. | 0.150 |
| **Metric** | dist.(C&W $\ell_2$) | 1.335 | 2.552 | 2.282 | 1.229 | 1.884 | 1.928 | 1.439 | 2.312 | 1.521 | 2.356 |
| | $\epsilon_{\min}$ ($\ell_\infty$) | 0.050 | 0.165 | 0.110 | 0.083 | 0.162 | 0.082 | 0.106 | 0.176 | 0.072 | 0.171 |
| | Image | | | | | | | | | | |
| **Weight** | $\ell_0$ | 0.481 | 0. | 0.378 | 0. | 0. | 0. | 0.352 | 0. | 0. | 0.648 |
| | $\ell_1$ | 0.690 | 0. | 0.310 | 0. | 0. | 0. | 0.093 | 0.205 | 0. | 0.701 |
| | $\ell_2$ | 0.589 | 0.069 | 0.208 | 0. | 0.134 | 0.064 | 0.260 | 0.077 | 0. | 0.600 |
| | $\ell_\infty$ | 0.864 | 0. | 0.084 | 0. | 0.052 | 0.079 | 0.251 | 0.156 | 0. | 0.514 |
| **Metric** | dist.(C&W $\ell_2$) | 2.267 | 1.656 | 2.053 | 1.359 | 0.861 | 1.733 | 1.967 | 1.741 | 1.031 | 2.413 |
| | $\epsilon_{\min}$ ($\ell_\infty$) | 0.171 | 0.088 | 0.143 | 0.117 | 0.086 | 0.100 | 0.097 | 0.096 | 0.038 | 0.132 |
| | Image | | | | | | | | | | |
| **Weight** | $\ell_0$ | 0. | 0. | 0.753 | 0. | 0.247 | 0. | 0. | 0. | 1.000 | 0. |
| | $\ell_1$ | 0.018 | 0. | 0.567 | 0. | 0.416 | 0.347 | 0. | 0. | 0.589 | 0.063 |
| | $\ell_2$ | 0. | 0. | 0.595 | 0. | 0.405 | 0.346 | 0. | 0. | 0.654 | 0. |
| | $\ell_\infty$ | 0. | 0. | 0.651 | 0. | 0.349 | 0.239 | 0. | 0. | 0.761 | 0. |
| **Metric** | dist.(C&W $\ell_2$) | 1.558 | 1.229 | 1.939 | 0.297 | 1.303 | 0.940 | 1.836 | 1.384 | 1.079 | 2.027 |
| | $\epsilon_{\min}$ ($\ell_\infty$) | 0.084 | 0.088 | 0.122 | 0.060 | 0.094 | 0.115 | 0.103 | 0.047 | 0.125 | 0.100 |

**Table A10:** Interpretability of domain weight $w$ for universal perturbation to multiple inputs on MNIST (*Digit 5 to 9*). Domain weight $w$ for different images under $\ell_p$-norm ($p = 0, 1, 2, \infty$) and two metrics measuring the difficulty of attacking single image are recorded, where dist. ($\ell_2$) denotes the the minimum distortion of successfully attacking images using C&W ($\ell_2$) attack; $\epsilon_{\min}$ ($\ell_\infty$) denotes the minimum perturbation magnitude for $\ell_\infty$-PGD attack.

| Image | | 5 | 5 | 5 | 5 | 5 | 5 | 5 | 5 | 5 | 5 |
|---|---|---|---|---|---|---|---|---|---|---|---|
| Weight | $\ell_0$ | 0. | 0.062 | 0.254 | 0. | 0.684 | 0.457 | 0. | 0. | 0.542 | 0. |
| | $\ell_1$ | 0.131 | 0.250 | 0. | 0. | 0.619 | 0.033 | 0.157 | 0.005 | 0.647 | 0.158 |
| | $\ell_2$ | 0.012 | 0.164 | 0.121 | 0. | 0.703 | 0.161 | 0.194 | 0. | 0.508 | 0.136 |
| | $\ell_\infty$ | 0.158 | 0.008 | 0.258 | 0. | 0.576 | 0.229 | 0.179 | 0. | 0.401 | 0.191 |
| Metric | dist. ($\ell_2$) | 1.024 | 1.532 | 1.511 | 1.351 | 1.584 | 1.319 | 1.908 | 1.020 | 1.402 | 1.372 |
| | $\epsilon_{\min}$ ($\ell_\infty$) | 0.090 | 0.106 | 0.085 | 0.069 | 0.144 | 0.106 | 0.099 | 0.0748 | 0.131 | 0.071 |
| Image | | 6 | 6 | 6 | 6 | 6 | 6 | 6 | 6 | 6 | 6 |
| Weight | $\ell_0$ | 0.215 | 0. | 0. | 0.194 | 0.590 | 0.805 | 0. | 0. | 0.195 | 0. |
| | $\ell_1$ | 0.013 | 0. | 0. | 0.441 | 0.546 | 0.775 | 0. | 0. | 0.225 | 0. |
| | $\ell_2$ | 0.031 | 0. | 0. | 0.410 | 0.560 | 0.767 | 0. | 0. | 0.233 | 0. |
| | $\ell_\infty$ | 0. | 0. | 0. | 0.459 | 0.541 | 0.854 | 0. | 0. | 0.146 | 0. |
| Metric | dist. ($\ell_2$) | 1.199 | 0.653 | 1.654 | 1.156 | 1.612 | 2.158 | 0. | 1.063 | 1.545 | 0.147 |
| | $\epsilon_{\min}$ ($\ell_\infty$) | 0.090 | 0.017 | 0.053 | 0.112 | 0.158 | 0.159 | 0.020 | 0.069 | 0.145 | 0.134 |
| Image | | 7 | 7 | 7 | 7 | 7 | 7 | 7 | 7 | 7 | 7 |
| Weight | $\ell_0$ | 0.489 | 0. | 0. | 0.212 | 0.298 | 0.007 | 0.258 | 0.117 | 0.482 | 0.136 |
| | $\ell_1$ | 0.525 | 0.190 | 0. | 0.215 | 0.070 | 0.470 | 0.050 | 0.100 | 0.343 | 0.038 |
| | $\ell_2$ | 0.488 | 0.165 | 0. | 0.175 | 0.172 | 0.200 | 0.175 | 0.233 | 0.378 | 0.014 |
| | $\ell_\infty$ | 0.178 | 0.263 | 0. | 0.354 | 0.205 | 0.258 | 0.207 | 0.109 | 0.426 | 0. |
| Metric | dist. ($\ell_2$) | 1.508 | 1.731 | 1.291 | 1.874 | 1.536 | 1.719 | 2.038 | 1.417 | 2.169 | 0.848 |
| | $\epsilon_{\min}$ ($\ell_\infty$) | 0.110 | 0.125 | 0.089 | 0.126 | 0.095 | 0.087 | 0.097 | 0.084 | 0.135 | 0.077 |
| Image | | 8 | 8 | 8 | 8 | 8 | 8 | 8 | 8 | 8 | 8 |
| Weight | $\ell_0$ | 0. | 0. | 1.000 | 0. | 0. | 0.246 | 0. | 0. | 0. | 0.754 |
| | $\ell_1$ | 0. | 0.180 | 0.442 | 0.378 | 0. | 0.171 | 0. | 0. | 0. | 0.829 |
| | $\ell_2$ | 0. | 0.298 | 0.593 | 0.109 | 0. | 0.330 | 0. | 0. | 0. | 0.670 |
| | $\ell_\infty$ | 0. | 0.377 | 0.595 | 0.028 | 0. | 0.407 | 0. | 0. | 0. | 0.593 |
| Metric | dist. ($\ell_2$) | 1.626 | 1.497 | 1.501 | 1.824 | 0.728 | 1.928 | 1.014 | 1.500 | 1.991 | 1.400 |
| | $\epsilon_{\min}$ ($\ell_\infty$) | 0.070 | 0.153 | 0.156 | 0.156 | 0.055 | 0.171 | 0.035 | 0.090 | 0.170 | 0.161 |
| Image | | 9 | 9 | 9 | 9 | 9 | 9 | 9 | 9 | 9 | 9 |
| Weight | $\ell_0$ | 1. | 0. | 0. | 0. | 0. | 0. | 0.665 | 0.331 | 0. | 0.004 |
| | $\ell_1$ | 0.918 | 0. | 0.012 | 0. | 0.070 | 0. | 0.510 | 0.490 | 0. | 0. |
| | $\ell_2$ | 0.911 | 0. | 0.089 | 0. | 0. | 0. | 0.510 | 0.490 | 0. | 0. |
| | $\ell_\infty$ | 0.935 | 0. | 0.065 | 0. | 0. | 0. | 0.665 | 0.331 | 0. | 0.004 |
| Metric | dist. ($\ell_2$) | 1.961 | 1.113 | 1.132 | 1.802 | 0.939 | 1.132 | 1.508 | 1.335 | 1.033 | 1.110 |
| | $\epsilon_{\min}$ ($\ell_\infty$) | 0.144 | 0.108 | 0.083 | 0.103 | 0.079 | 0.041 | 0.090 | 0.103 | 0.083 | 0.044 |

