# OpenReview forum: "Towards A Unified Min-Max Framework for Adversarial Exploration and Robustness"
_ICLR.cc/2020/Conference — Reject_

### Official Review · AnonReviewer2 · 2019-10-25
**Official Blind Review #2**

**Rating:** 8

**Review:**

The paper presents a unified framework for adversarial training & robustness. The problem is important and interesting. The proposed framework has solid theory and is well conceived. A generic method is proposed with O(1/T) convergence rate, which is also empirically demonstrated with good performance on often-used MNIST and CIFAR-10 benchmarks. An alternating multi-step PGD is also proposed. Empirical experiments are thorough and well organized. Overall I feel it is a well written paper with sufficient contributions and is of interest to a range of ICLR audience.


**Experience Assessment:**

I have read many papers in this area.

**Review Assessment: Checking Correctness Of Derivations And Theory:**

I assessed the sensibility of the derivations and theory.

**Review Assessment: Checking Correctness Of Experiments:**

I assessed the sensibility of the experiments.

**Review Assessment: Thoroughness In Paper Reading:**

I read the paper at least twice and used my best judgement in assessing the paper.

---

> ### Author Response · Authors · 2019-11-11
> **Response to AnonReviewer2**
>
> We thank Reviewer #2 very much to recognize our contributions and have positive comments on our paper! We will further update our paper by adding new experiment results (suggested by other reviewers) as well as improving our presentation.

---

### Official Review · AnonReviewer3 · 2019-10-28
**Official Blind Review #3**

**Rating:** 3

**Review:**

Note: I applied a higher standard to this paper given that it significantly exceeds the recommended page limit. Furthermore, important details are left out in the appendices, which make it difficult to read the main body of the paper in a self-contained fashion. Given that the main body was already over the recommended page limit, I did not read the appendices.

This paper generalizes the min max formulation of adversarial training, and proposes a formulation that encompasses adversarial training of an ensemble, robustness to universal adversarial examples, and robustness to non-adversarial transformations. This formulation is used to derive an adversarial training procedure that trains against the worst-case adversarial example among adversarial examples generated by a set of attacks. Experiments seek to demonstrate applicability of this framework to both attacks and defenses.

As far as experiments are concerned, Section 4.1 presents results on MNIST, which is known to be a poor dataset to study adversarial examples on [https://arxiv.org/abs/1902.06705]. If models C and D are more difficult to attack, could better baselines be employed than attacking the ensemble A+B+C+D? For instance, would an adversary evading models C+D only perform better? It is difficult to draw insights that are generally applicable from a single ensemble. How was the ensemble chosen? Why would a defender add models which are known to be significantly less robust to the ensemble?

When discussing universal perturbations, how are they generated? Given that the performance of the proposed approach significantly degrades average evasion across all images from all groups, what is the threat model for an adversary being interested in group-level success rather than average evasion across all images from all groups? How were the values of K chosen? This comment also 	applies to experiments over data transformations. For these experiments, what was the value of K?

As far as the defensive perspective is concerned, it is not clear whether the improvements observed are statistically significant. Were multiple runs averaged to produce Table 4? Given that without DPAR, the improvement is negligible, this is important to interpret results. It appears that most of the robustness gains in both the average and max settings stem from DPAR. This should be clearly surfaced in the introduction and presentation of contributions if DPAR is required for the proposed generalized min max formulation to improve robustness. In particular, it is not clear whether DPAR is “a beneficial supplement to adversarial training” or a required supplement to adversarial training - per the formulation in this paper.


There are issues with grammar throughout the document, which make it difficult to read. Some specific issues:

1 - Adversarial attack is a tautology (an attack is always adversarial)

7 - What does “robust” adversarial attack mean?

7 - What is CAAD-18?

7 - Define ASR_all: what does evade mean here? Is the attack targeted or untargeted?

7 - What is an “advanced” DNN?

**Experience Assessment:**

I have published one or two papers in this area.

**Review Assessment: Checking Correctness Of Derivations And Theory:**

I assessed the sensibility of the derivations and theory.

**Review Assessment: Checking Correctness Of Experiments:**

I carefully checked the experiments.

**Review Assessment: Thoroughness In Paper Reading:**

I read the paper at least twice and used my best judgement in assessing the paper.

---

> ### Author Response · Authors · 2019-11-11
> **Response to AnonReviewer3 (Part 1/3)**
>
> Thanks for your valuable comments and suggestions.
>
> Q1: The paper exceeds the recommended page limit (so apply higher standard). It is not self-contained in the main body and leaves some important details in the appendix.
> A1: Thank you. The reviewer’s suggestion reminds us to better organize the paper and to make our presentation clearer. For ease of understanding, we will add a table of contents in the Appendix for ease of associating our main sections with supplementary details in Appendix. Moreover, in the main paper (especially in the experiment section), we will make a better organization when the figures/tables/sections in Appendix are cited.
>
> Q2: Sec 4.1 presents results on MNIST, which is known to be a poor dataset to study adversarial examples (https://arxiv.org/abs/1902.06705)
> A2: First, we conducted the same experiments on CIFAR-10 and tested more models (e.g., VGG16, Wide-ResNet, GoogLeNet) in the appendix (Table 3, 4). To make it more clearly, we will move the CIFAR-10 experiment to the main body in the updated paper. Moreover, we obtained consistent attack results on both MNIST and CIFAR-10.
>
> Second, MNIST is a dataset, which provides images of easy visualization. For example, we use MNIST to visualize the effect of self-adjusted weights $\mathbf w$ on the attack performance (Table A8 and A9). It is clear to see that the larger domain weights correspond to the MNIST letters with clearer appearance, implying that the learnable weights $\mathbf w$ could offer visual interpretability of “image robustness”.
>
> Third, we hesitate to call MNIST a poor dataset for studying adversarial examples. To the best of our knowledge, many works (Madry et al, 2017; Athalye et al., 2019; Tramèr et al., 2019) considered MNIST as a standard dataset. Even for the seminal work (Carlini et al., 2019), we did not see a clear objection on MNIST to study adversarial examples. A possible relevant point in the provided paper is that one should consider different perturbation radiuses for different datasets; for example, $\epsilon=0.2$ ($\ell_\infty$ attack) used for MNIST becomes too large for CIFAR-10. In our work, we follow the commonly-used setting of the perturbation radius, 0.2 for MNIST, 0.05 or 0.03 for CIFAR-10.
>
> Reference:
> Towards Deep Learning Models Resistant to Adversarial Attacks. Madry et al., ICLR 2017.
> Obfuscated Gradients Give a False Sense of Security: Circumventing Defenses to Adversarial Examples. Athalye et al., ICLR 2019.
> Adversarial Training and Robustness for Multiple Perturbations. Tramèr et al., NeurIPS 2019.
> On Evaluating Adversarial Robustness. Carlini et al., 2019.
>
> Q3: Questions about ensemble models (better baselines, insights of attacking model ensembles).
> A3: Thanks for the suggestion.
>
> First, we would like to highlight that the weights $\mathbf w$ to combine ensemble models are learnable, which avoids supervised manual adjustment. Thus, our approach does not need prior knowledge on the robustness level of different models from both attacker and defender’s perspectives.
>
> Second, following the reviewer's suggestion, we conduct additional experiments (Table 1 in https://tinyurl.com/t6hax2m) to show that it is nontrivial to set the heuristic weights beforehand. For example, when an adversary evades models C+D only, the generated adversarial examples have a poor transferability to even less robust models A and B. This implies that having the ensemble attack to learn the adjusted weights by itself not only avoids the issue of heuristic weight selection but also boosts the transferability of attacks to different models. Note that it is actually a common practice to attack ensemble models to promote the transferability to unknown black-box models. For instance, in NIPS 2017, CAAD-2018 competitions, the winner solutions usually integrate multiple adversarially trained models (usually > 5) to enhance the transferability with equal or manually fine-tuned weights. However, we have shown in Table1 that our min-max solution outperforms this averaging strategy.
>
> Third, our approach does not rely on specific choices of the model ensemble. For available public models, the importance weights $\mathbf w$ are jointly learned, and the resulting results of $\mathbf w$ could be used as metrics to reveal the robustness of individual models in the ensemble (Figure 1c).

---

> > ### Author Response · Authors · 2019-11-11
> > **Response to AnonReviewer3 (Part 2/3)**
> >
> > Q4: Questions on  universal perturbations.
> > A4:  First, the universal perturbation is generated by solving the problem (6) using APGD, where we set $\gamma = 4$. It is worth noting that as $\gamma$ approaches 0 and $\infty$, problem (6) reduces the maximum and the averaging strategy for generating universal perturbation.
> >
> > Second, our goal is not to propose a new threat model for an adversary being interested in group-level success. Instead, we consider group-level success since it is a complementary metric to average-evasion success:  A high average-evasion success rate does not imply a high group-level success and vice versa. We would like to show that as K increases, attacking all of the images in an entire group becomes much more difficult. As a result, the proposed min-max attack can achieve better group-level success but possibly at the cost of degraded average-evasion success. As K is small, the min-max strategy outperforms the averaging strategy under both group-level success and average-evasion success (e.g., K = 2 for MNIST, and K = 2 or 4 for CIFAR). In the revision, we will make our motivation on group-level success and our comparison with average-evasion success clearer.
> >
> > Third, in the example of universal perturbation, we compare the min-max strategy and the averaging strategy by choosing K=2, 4, 5, 10, which are factors that can be divided by the total number of images 10,000. In the example of data transformation, we choose K = 6 data transformations following previous works (Athalye et al., 2018; Athalye et al., 2019;  Brown et al., 2017), i.e., flipping horizontally (flh) or vertically (flv), adjusting brightness (bri), performing gamma correction (gam) and cropping (crop) (see Table 3).
> >
> > Reference:
> > Synthesizing robust adversarial examples. Athalye et al., ICLR 2018.
> > Obfuscated gradients give a false sense of security. Athalye et al., ICLR 2019.
> > Adversarial patch. Brown et al., 2017.
> >
> > Q5: not clear whether the improvements observed for defense are statistically significant (were multiple runs averaged in Table 4, without DPAR, the improvement is negligible)
> > A5: The improvements are statistically significant - the experiment is repeated ten times under different random seeds (see Figure A3b for the learning curve).
> >
> > In this setting ($\epsilon_{\ell_\infty} = 0.2$, $\epsilon_{\ell_2} = 2.0$), the improvement of min-max over average strategy is small because the large overlapping between $\ell_\infty$ and $\ell_2$ balls (see Sec 4.2 and Figure A2). That is also the reason why promoting diversity helps improve robustness further. Figure 3a also demonstrates that the learned weights for the two types of perturbations are very close. Moreover, Figure A3a shows that our proposed min-max scheme results in faster convergence than the averaging scheme due to the benefit of self-adjusted domain weights.
> >
> > Also, we do not think that 1% improvement is negligible since it is achieved over multiple runs and different scenarios. Moreover, the robustness improvement of AT under a single perturbation type by the ICML’19 work [Table 1, Wang et al. 2019] is also around 1%, although a different network architecture and an AT method were considered in our work.
> >
> > Reference:
> > On the Convergence and Robustness of Adversarial Training. Wang et al., ICML 2019.
> >
> > Q6: should emphasize the contribution of DPAR (whether a beneficial or required supplement to adversarial training)
> > A6: Thanks for the suggestion. DPAR is a beneficial supplement to AT under multiple types of $\ell_p$ perturbations. In Table 4, we observe that DPAR yields consistent robustness improvement under both Acc$_{\mathrm{adv}}^{\mathrm{avg}}$ and  Acc$_{\mathrm{adv}}^{\mathrm{max}}$ at different scenarios. We will add more discussions in the revised manuscript.
> >
> > Q7: Some typos in the paper
> > A7: Thanks for the meticulous proofreading. We will correct these typos and further revise our manuscript.

---

> > > ### Author Response · Authors · 2019-11-11
> > > **Response to AnonReviewer3 (Part 3/3)**
> > >
> > > Detailed comments & questions:
> > > 1 - Adversarial attack is a tautology (an attack is always adversarial)
> > > “Adversarial attack” is a terminology that is widely used in the community (Huang et al., 2017; Madry et al., 2018; Samangouei et al., 2018). It usually represents well-crafted prediction-evasion attacks during the inference. This is in contrast with the training-phase data poisoning attack (Steinhardt et al., 2017; Shafahi et al., 2018).
> > >
> > > Reference:
> > > Adversarial Attacks on Neural Network Policies. Huang et al., ICLR 2017.
> > > Towards Deep Learning Models Resistant to Adversarial Attacks. Madry et al., ICLR 2018.
> > > Defense-GAN: Protecting Classifiers Against Adversarial Attacks Using Generative Models. Samangouei et al., ICLR 2018.
> > > Certified Defenses for Data Poisoning Attacks. Steinhardt et al., NeurIPS 2017.
> > > Poison Frogs! Targeted Clean-Label Poisoning Attacks on Neural Networks. Shafahi et al., NeurIPS 2018.
> > >
> > > 7 - What does “robust” adversarial attack mean?
> > > The “robust” refers to the improvement on “worst-case” attack performance over multiple domains. We will try to make this point clearer and more accurate.
> > >
> > > 7 - What is CAAD-18?
> > > As shown in Sec 4.1, CAAD-18 denotes “Competition on Adversarial Attacks and Defenses 2018”.
> > >
> > > 7 - Define ASR$_{all}$: what does evade mean here? Is the attack targeted or untargeted?
> > > ASR$_{all}$ refers to the attack success rate (ASR) of fooling model ensembles simultaneously. Our framework is applicable to both untargeted and targeted attacks. In the experiments, we focus on the former setting. Although this metric was defined in the paper, the reviewer’s comment reminds us to have a clearer organization of our experiment details. Thanks!
> > >
> > > 7 - What is an “advanced” DNN?
> > > Here, “advanced” DNN means more complicated DNNs such as VGG, ResNet, Wide-ResNet, GoogLeNet. We will clarify it in the revision.

---

### Official Review · AnonReviewer4 · 2019-11-04
**Official Blind Review #4**

**Rating:** 3

**Review:**

The paper studies how a min-max framework can incorporate different tasks related to adversarial robustness. Specifically, the authors study adversarial attacks against model ensembles, universal perturbations, and attacks constrained by the union of Lp norms. They propose optimizing a probability distribution over "domains" (models in an ensemble, inputs, Lp balls; respectively per task) and regularizing it to be close to uniform. They perform experiments to evaluate their method.

From a conceptual point of view, I did not find the contribution of the paper significant. All of the tasks discussed are direct application of the min-max framework and have been studied to a certain extent in prior work (https://arxiv.org/abs/1811.11304, https://arxiv.org/abs/1706.04701, https://arxiv.org/abs/1904.13000). The novel tools introduced are the regularizer on simplex probability and the attack diversity regularizer. However, the theoretical justification for these tools is rather weak and their utility would need to be demonstrated empirically.

From an experimental point of view, the baselines considered are very weak. At a high level, the authors compare their version of min-max optimization to a very simple average-case optimization. In order to demonstrate the utility of the tools introduced the authors would need to at least compare to a reasonable min-max baseline. For instance, a very simple heuristic capping the loss of each domain in the finite-sum formulation (https://arxiv.org/abs/1811.11304) would be the bare minimum. In their current state, the experiments only demonstrate that a min-max approach outperforms an average case approach, which is fully expected. At the same time, the diversity regularizer does seem to offer some empirical gains and I would encourage the authors to investigate further.

Overall, the conceptual and experimental contributions of the paper are rather weak and I thus recommend rejection.

=========================
UPDATE: I appreciate the authors' response and additional experimental results.

I am still quite concerned about the universal perturbation baseline. I suspect that the clipping factor used might be too large since clipping barely has any impact (the attack is still focusing too much on B ignoring C). Conceptually, clipping should be quite similar to a min-max formulation.

I do see that the proposed method outperforms the one proposed in Shafahi et al in terms of universal adversarial training. I feel like this is a more reasonable baseline and I am increasing my score to a 3.

**Experience Assessment:**

I have published in this field for several years.

**Review Assessment: Checking Correctness Of Derivations And Theory:**

I assessed the sensibility of the derivations and theory.

**Review Assessment: Checking Correctness Of Experiments:**

I carefully checked the experiments.

**Review Assessment: Thoroughness In Paper Reading:**

I read the paper thoroughly.

---

> ### Author Response · Authors · 2019-11-11
> **Response to AnonReviewer4 (Part 1/2)**
>
> Thanks for your valuable comments and suggestions.
>
> Q1: Conceptual novelty?
>
> A1: Thanks for your comments. In the revision, we will clarify our conceptual contributions, and elaborate on our differences with the existing works (Shafahi et al., 2018; Tramèr et al., 2019; He et al., 2017).
>
> We are sorry to hear that “All of the tasks discussed are direct applications of the min-max framework and have been studied to a certain extent in prior work”. We strongly believe that our work contains substantial differences to the existing works, and is not a direct application of the min-max framework.
>
> Differences to  (Shafahi et al., 2018; Tramèr et al., 2019; He et al., 2017).
> The prior work (Shafahi et al., 2018)  proposed a min-max based AT by leveraging universal perturbations (rather than per-image perturbation). We highlight two main differences even just at the defense side. First, different from Shafahi et al., we generalize AT subject to mixed types of $\ell_p$ adversarial perturbations. Figure A2 motivates on why the consideration of multiple perturbation types could matter in AT. Second, our min-max formulation (11) stems from the min-max-max formulation (18), where the last max step is conducted on the importance weights $\mathbf w$. In Lemma 1, it is our contribution to show the equivalence between (18) and (11).
>
> Although the prior work (Tramèr et al., 2019) considered a modified AT method subject to multiple types of adversarial perturbations, our work is still quite different from it. Note that some differences had been highlighted in Sec. 2 of the original manuscript. We make further clarification as below. First, the incorporation of domain weights $\mathbf w$, the corresponding regularization on $\mathbf w$ and the diversity-promoting regularization are new to AT under multiple types of adversarial perturbations. Second, our proposed framework is general, which applies to both attack generation and AT. Even at the defense side, our approach covers the formulations in Tramèr et al., as $\gamma$ approaches $0$ and $\infty$. Third, please note that ours and Tramèr et al., 2019 are actually independent works.
>
> The prior work (He et al., 2019) considered an ensemble-based defensive method. We feel that this is less relevant to the min-max AT framework. Our differences to the previous work (Shafahi et al., 2018) hold for He et al., 2019.
>
> A summary of our conceptual contributions.
>
> (Attack) Even if the concept of min-max optimization was used in other works, our formulation on the min-max attack and its specification to ensemble attack, universal perturbation, and attack under physical transformations are new to the field. In Proposition 1, we derived the analytical solution to the projection operator subject to the intersection of $\ell_p$ norm ($p = 0,1,2,\infty$) inequality constraint and the box constraint. This is different from the conventional attack design, where the  $\ell_p$ norm was regularized in the objective function. The derived solution to the projection operator subject to the hard constraints also facilitates the implementation of AT under multiple types of adversarial perturbations (see Sec. 3.2).
> (Defense) The proposed min-max formulation (11) on AT under mixed types of $\ell_p$ perturbations is not trivial. It actually stems from the min-max-max formulation (18), where the last max step is conducted on the importance weights $\mathbf w$. In Lemma 1, we show that problem (18) can equivalently be transformed into the proposed min-max problem (11). Moreover, the diversity regularization on multiple perturbation directions is new to AT.
> In both a) and b), the introduction of self-adjusted weights $\mathbf w$ and the strongly concave regularization are not trivial. First, the learnable $\mathbf w$ can adjust the model robustness or attack power automatically during the training. Second, the introduction of strongly concave regularization is useful, which ensures $O(1/T)$ convergence rate for APGD (Theorem 1) and helps the training process in AT under multiple types of $\ell_p$ perturbations (Figure A3).
>
> Reference:
> Universal Adversarial Training. Shafahi et al., 2018.
> Adversarial Training and Robustness for Multiple Perturbations. Tramèr et al., NeurIPS 2019.
> Adversarial Example Defenses: Ensembles of Weak Defenses are not Strong. He et al., WOOT 2017

---

> > ### Author Response · Authors · 2019-11-11
> > **Response to AnonReviewer4 (Part 2/2)**
> >
> > Q2: Experimental contributions and baselines?
> >
> > A2:  We would like to thank the reviewer’s suggestion on baselines and additional empirical studies.
> >
> > First, we summarize the newly conducted experiments (https://tinyurl.com/t6hax2m).
> >
> > To demonstrate the usefulness of the self-adjusted domain weights $\mathbf w$, we compare the performance of the proposed ensemble attack with that of the min-max approach using a heuristic weighting structure as well as a clipping loss strategy used in Shafahi et al., 2018. We show that the use of learnable $\mathbf w$ avoids supervised manual adjustment on the importance of attack losses, and our approach outperforms the heuristic weighting structure  (Table 1 in https://tinyurl.com/t6hax2m). We also conduct a sensitivity analysis of $\gamma$ and show how the proposed regularization affects the eventual performance (Figure 2 in https://tinyurl.com/t6hax2m).
> >
> > We add Universal Adversarial Training (UAT) (Shafahi et al., 2018.) as one of our defense baselines in Table 2 (https://tinyurl.com/t6hax2m). We find that a) our proposed approach outperforms UAT under per-image $\ell_p$ attacks. Taking Table 2a as an example, our avg and min-max generalized AT (with DPAR) result in average 17.85% and 17.97% improvement in adversarial test accuracy (ATA), b) our approach has just 3.7% degradation in ATA when encountering universal attacks, and c) both methods yield very similar normal test accuracy.
> >
> > Second, we do not think that our previous experimental contributions are weak. We would like to highlight the following contributions.
> >
> > We conducted extensive experiments on showing the power of our proposed min-max formulation and method on the design of adversarial attacks (Figures 1, Table 1-3, Table A3-A6). We also provided empirical insights on why the self-adjusted weights $\mathbf w$ matter (see Figures 1c, 3a, Table A8). For example, the min-max universal perturbation offers interpretability of “image robustness” by associating domain weights with image visualization:  Larger domain weights correspond to the MNIST letters with clearer appearance; see Table A8 and A9 (Appendix D.3).
> > Besides the defensive results that we showed in Sec. 4.2, we conducted additional experiments in the Appendix. For example, in Figure A2, we provide insights on why AT is trained over multiple perturbation types could matter: It leads to a defense consistently strong through different $\ell_p$ attacks (see Table 4, Figure 3b-3c, Figure A7). In Figure A3, we show that our proposed defensive method leads to a faster convergence during training compared to average-based AT, especially when the strengths of two attacks diverge greatly.
> >
> > Reference:
> > Universal Adversarial Training. Shafahi et al., 2018.

---

### Author Response · Authors · 2019-11-11
**General Response to Reviewers**

We thank the reviewers for their valuable comments and suggestions. We have made our best efforts to address all reviewers' concerns and questions. Some highlights are provided below:

1) We clarified the novelty of our work and carefully made a distinction between ours and others suggested by reviewers. We would like to respectfully mention that the seemingly 'direct' application of min-max optimization does not mean that our contributions are minor, instead it demonstrates the generalizability and flexibility of the proposed approach in both attack design and robust training. And we have made extensive experiments (the additional experiments suggested from reviewers are quite valuable) to support our conceptual contributions. The detailed response can be found when answering each specific question.

2) We conducted new experiments as suggested by reviewers and the results are available in this anonymous link: https://tinyurl.com/t6hax2m. Specifically, we added 1) stronger heuristic baselines with the prior knowledge of model robustness (Table 1); 2) comparison with universal adversarial training (Shafahi et al., 2018); 3) sensitivity analysis of proposed quadratic regularizer on probability simplex.

We will update the manuscript soon based on reviewers' feedback. Any further discussions and suggestions are highly welcome and appreciated.

Reference:
Universal Adversarial Training. Shafahi et al., 2018.

---

### Author Response · Authors · 2019-11-13
**Revised manuscript has been uploaded**

We would like to thank the reviewers again for their thoughtful reviews and valuable comments. We have made our best efforts to improve the manuscript according to the comments. The key changes are listed as follows:

- clarifying the novelty and contributions
- adding heuristic weighting schemes baselines (Appendix D2, Table A4)
- comparing proposed generalized AT with another min-max baseline - universal adversarial training (Shafahi et al., 2018) (Appendix E2, Table A7)
- conducting a sensitivity analysis of quadratic regularizer on probability simplex (Appendix D4, Figure A1)
- improving the presentation of the paper, adding a table of contents in Appendix for ease of associating our main sections with supplementary details

Any further discussions are highly welcome and appreciated!

Reference:
[1] Universal Adversarial Training. Shafahi et al., 2018.

---

### Decision · Program_Chairs · 2019-12-19

**Decision:**

Reject

**Comment:**

This submission studies an interesting problem. However, as some of the reviewers point out, the novelty of the proposed contributions is fairly limited.